# Closed-loop auditory stimulation method to modulate sleep slow waves and motor learning performance in rats

Carlos G Moreira[1], Christian R Baumann[1,2,3], Maurizio Scandella[1],
Sergio I Nemirovsky[4], Sven Leach[5], Reto Huber[2,3,5,6], Daniela Noain[1,2,3]*

[1]Department of Neurology, University Hospital Zurich, University of Zurich, Zurich, Switzerland; [2]University Center of Competence Sleep & Health Zurich (CRPP), University of Zurich, Zurich, Switzerland; [3]Neuroscience Center Zurich (ZNZ), Zurich, Switzerland; [4]Institute of Biological Chemistry, School of Exact and Natural Sciences (IQUIBICEN). CONICET – University of Buenos Aires, Buenos Aires, Argentina; [5]Child Development Center, University Children's Hospital Zurich, University of Zurich, Zurich, Switzerland; [6]Department of Child and Adolescent Psychiatry and Psychotherapy, Psychiatric Hospital, University of Zurich, Zurich, Switzerland

**Abstract** Slow waves and cognitive output have been modulated in humans by phase-targeted auditory stimulation. However, to advance its technical development and further our understanding, implementation of the method in animal models is indispensable. Here, we report the successful employment of slow waves' phase-targeted closed-loop auditory stimulation (CLAS) in rats. To validate this new tool both conceptually and functionally, we tested the effects of up- and down-phase CLAS on proportions and spectral characteristics of sleep, and on learning performance in the single-pellet reaching task, respectively. Without affecting 24 hr sleep-wake behavior, CLAS specifically altered delta (slow waves) and sigma (sleep spindles) power persistently over chronic periods of stimulation. While up-phase CLAS does not elicit a significant change in behavioral performance, down-phase CLAS exerted a detrimental effect on overall engagement and success rate in the behavioral test. Overall CLAS-dependent spectral changes were positively correlated with learning performance. Altogether, our results provide proof-of-principle evidence that phase-targeted CLAS of slow waves in rodents is efficient, safe, and stable over chronic experimental periods, enabling the use of this high-specificity tool for basic and preclinical translational sleep research.

*For correspondence:
daniela.noain@usz.ch

Competing interest: The authors declare that no competing interests exist.

## Introduction

Noninvasive neuromodulation strategies are en vogue for their unique potential to diagnose and treat neurological and psychiatric disorders, or to rebalance the activity in dysfunctional brain networks. In particular, modulation of slow wave sleep (SWS) has increasingly gained attention over recent years. Tailoring novel stimulation tools to effectively optimize SWS profiles appears crucial to intervene SWS-related cognition (*Mander et al., 2013*; *Papalambros et al., 2017*), as well as other important brain and body processes, in both rodents and humans. Therefore, innovative approaches to further comprehend and eventually enable therapeutic implementations of SWS modulation have been tested (*Marshall et al., 2006*; *Massimini et al., 2007*; *Vyazovskiy et al., 2009*). More recently, auditory stimulation during SWS was successfully implemented in human subjects in laboratory-based settings. At first, the method disregarded the phase of the ongoing oscillatory activity in the brain (*Ngo et al., 2013b*; *Tononi et al., 2010*), but was later developed further to deliver auditory stimulation in synchrony with the brain's own rhythm in a closed-loop manner (*Ngo et al., 2013b*). Targeting

ongoing slow waves in their up-phase enhanced slow oscillations during SWS, while targeting the waves' down-phase had the opposite effect. The success of the human implementation of CLAS further encouraged the development of portable devices enabling acoustic stimulation in a home-based environment (*Ferster et al., 2019*).

Memory formation is one of the most intriguing sleep-dependent brain processes for which the links between SWS—characterized by brain slow oscillatory activity in the delta-frequency range (0.5–4 Hz) and oscillations <1 Hz among its most distinctive features in the human electroencephalogram (EEG) (*Timofeev, 2011*)—and motor learning (*Fischer et al., 2002*; *Stickgold, 2005*; *Walker et al., 2002*), synaptic downscaling (*Tononi and Cirelli, 2014*), and consolidation mechanisms (*Diekelmann and Born, 2010*; *Rasch and Born, 2013*) have been exhaustively studied. In this context, it was not surprising that the novel specific auditory modulation of SWS methods implemented in humans was first assessed in their capacity to interact with memory performance. Initially, Ngo and colleagues found enhanced performance during a paired-associates learning task in subjects undergoing up-phase auditory stimulation (*Ngo et al., 2013a*), and soon several other studies employing similar stimulation paradigms replicated the enhanced memory effect, although with more modest effect sizes (*Leminen et al., 2017*; *Ong et al., 2016*; *Papalambros et al., 2017*). In a novel implementation of this concept, some of us found that auditory disruption of slow waves perturbing local SWS in the motor cortex attenuated the brain's SWS-dependent capacity to undergo neuroplastic changes (*Fattinger et al., 2017*), corroborating that SWS integrity is critical to maintain cognitive efficiency. However, some authors proposed that enhancing slow waves and spindles by auditory stimulation was insufficient to improve memory above sham (*Cox et al., 2014*; *Ngo et al., 2019*; *Weigenand et al., 2016*), suggesting that the effects of auditory stimulation on memory are moderate and highly variable inter- and intra-studies, and evidencing that a better understanding of the method is needed.

Therefore, although a highly promising tool, the precise effects exerted by phase-targeted auditory stimulation at both electrophysiological and behavioral levels are not understood well enough yet. Moreover, effects of long-term interventions, as well as parameters' optimization, remain to be addressed. These limitations can partly be attributed to the lack of a closed-loop auditory stimulation (CLAS) paradigm in animals to facilitate both basic and preclinical research. For the first time, we explored here the effects exerted by phase-targeted CLAS of slow waves on slow-wave activity (SWA) in healthy rats. We conceptualized that up-phase CLAS will boost delta power during non-rapid eye movement (NREM) sleep, the sleep state rich in slow waves in animals, while down-phase CLAS will have the opposite effect. As a functional readout of CLAS of slow waves, we evaluated learning performance in the single-pellet reaching task (SPRT).

## Results

### CLAS' precision and phase detection

To study the effect of CLAS of slow waves in healthy rats (*Figure 1*, *Figure 1—figure supplement 1*), we recorded animals during an undisturbed 24 hr baseline (BL) period and, thereafter, delivering up-phase, mock, or down-phase auditory stimulation continuously for 16 days, while undergoing behavioral training in the SPRT (*Figure 2*). Online NREM sleep staging is a crucial step during CLAS of slow waves. Briefly, our online NREM sleep staging tool consists of three real-time decision features: root mean square (rms) of delta and beta frequency bands, and electromyogram (EMG). In a separate cohort (14 animals: up-phase [n = 5], mock [n = 4], or down-phase [n = 5]), we assessed the performance of our online NREM sleep staging process, as well as the accuracy of our phase detector. Overall, online NREM sleep staging had a mean sensitivity of 12%, indicating how strictly the staging tool identifies sustained NREM sleep, and specificity of 98%, highlighting the high rejection rate of events in disagreement with NREM characteristics (*Figure 3a*, see detailed methods 'EEG/EMG offline scoring and online staging validation' section). Regarding online precision of NREM sleep staging, 70% of epochs on average were confirmed offline as NREM sleep, whereas 18% were later identified as wakefulness and 2% as REM sleep (*Figure 3b*). Artefacts, i.e. EEG/EMG perturbations related to environmental interference rather than changes in brain state, constituted roughly 10% of epochs labelled online as NREM sleep. Phase histograms of a different cohort showed high accuracy in trigger distribution relative to the intended targets of 65° and 270° for up- and down-phase stimulation, respectively. Although different phases were targeted in this cohort, the phase detector is, in its

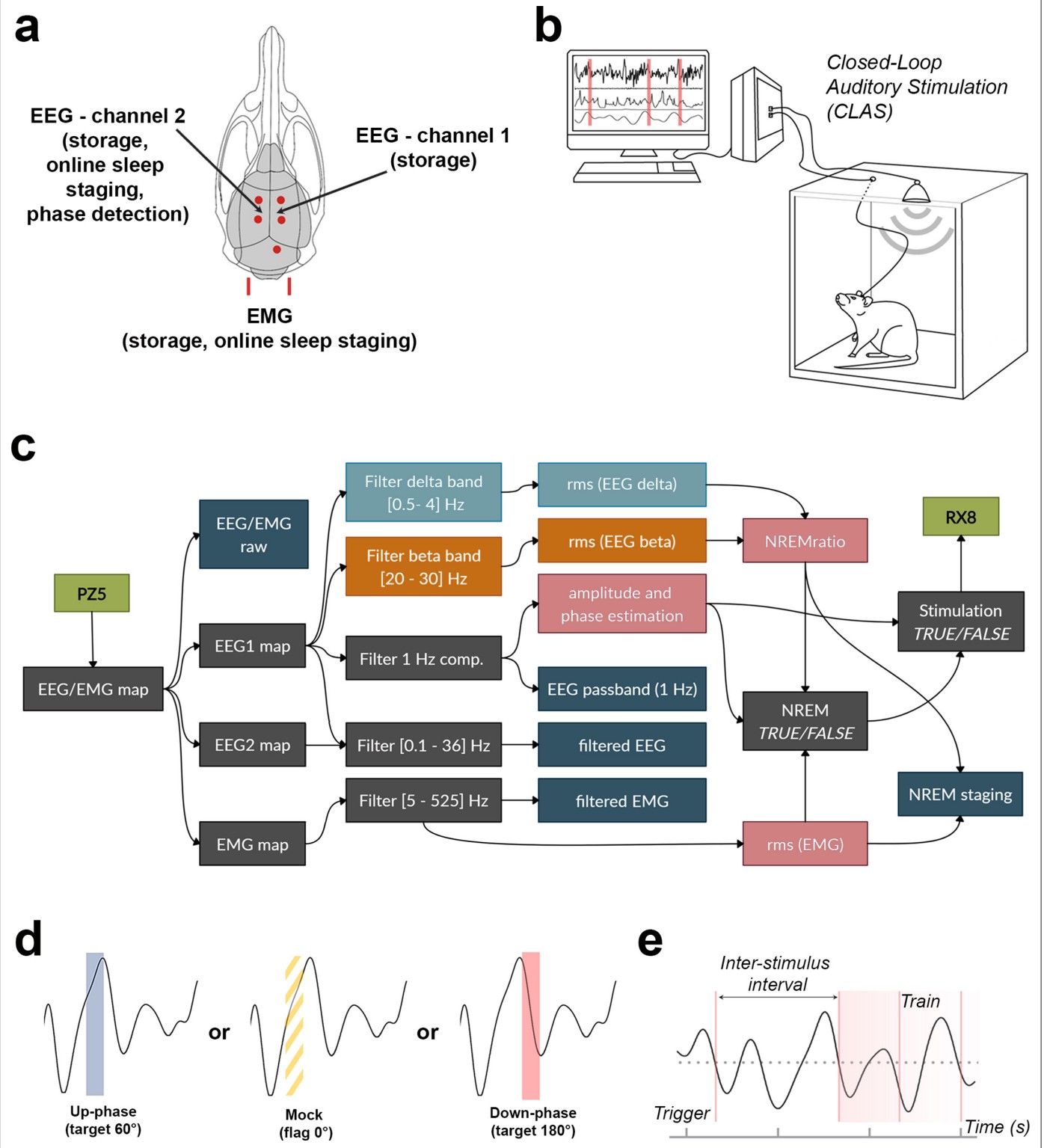

**Figure 1.** Schematic representation of electroencephalogram/electromyogram (EEG/EMG) headpiece, closed-loop auditory stimulation (CLAS) concept, and computational pipeline. (**a**) Dorsal skull schematic of the two subdural (differential) channels for cortical EEG and one nuchal EMG. (**b**) Closed-loop auditory stimulation setup: the system can accommodate up to ieight subjects in the same machine. (**c**) Processing tree for online non-rapid eye movement (NREM) staging and phase-targeted auditory stimulation. EEG/EMG is sampled and amplified via PZ5 NeuroDigitizer preamplifier (TDT, USA). Left: EEG (EEG1) is filtered in the delta and beta bands, and between 0.1 and 36 Hz for offline analysis; EMG signal is filtered between 5 and 525 Hz for offline analysis. Power estimations (rms(EEG delta) in orange, rms(EEG beta) in blue, and rms(EMG) in pink) and amplitude of EEG 1 Hz

*Figure 1 continued on next page*

*Figure 1 continued*

component form the basis for detection of NREM. When NREM is identified, auditory triggers are delivered phase-locked to the EEG 1 Hz component (RX8 MULTI-I/O processor). Intermediary operations are in gray and outputs for offline analysis are in dark blue. (d) Animals were stimulated with either up-phase (60°), mock stimulation (0°, sound muted by disconnection of the speaker), or down-phase (180°) CLAS. (e) Distribution of sound triggers was analyzed in terms of trains of triggers (count of any size sequences of triggers 1 s or less apart) and interstimulus interval (ISI, time between triggers). rms: root mean square.

The online version of this article includes the following figure supplement(s) for figure 1:

**Figure supplement 1.** Illustration of a sound-insulated chamber, online signal processing workflow, and SYNAPSE's interface.

core and performance, the same. Up-phase paradigm revealed *M* = 74.97° (SD = 38.10°; Var = 12.67°) while down-phase stimulation indicated a mean target *M* = 280.55° (SD = 49.67°; Var = 19.75°). In mock animals, we confirmed onset of muted triggers around the positive peak of the slow wave (*M* = 101.20°; SD = 46.89°; Var = 19.19°) (*Figure 3c*). The difference in variances reflects the gradual loss of accuracy when higher phases are being targeted, an intrinsic characteristic of our phase-detection feature. Additionally, we explored the distribution of triggers across sleep stages and observed that up-phase-stimulated animals received more sound triggers during NREM sleep than down-phase-stimulated subjects (up-phase: M = 4959, SD = 3063; down-phase: M = 3148, SD = 2719; *Figure 3—figure supplement 1a*). Nonetheless, the proportion of triggers did not differ between stimulation groups for any of the sleep stages (*Figure 3—figure supplement 1b*). The breakdown of CLAS triggers through pre- and motor-training phases shows that NREM sleep targeting by day (number and proportion) remained constant within each condition (*Figure 3—figure supplement 1c and d*).

## CLAS modulates delta and sigma power while preserving sleep proportions over 24 hr

We analyzed sleep proportions at BL and all 16 protocol days under CLAS (pre-, motor-, and non-training phases). Neither up-phase nor down-phase stimulation disturbed total NREM amount compared to mock condition (two-way repeated measures analysis of variance [RM-ANOVA], *time * condition* interaction, F(32, 215) = 0.929, p=0.582, *Figure 4a*). Moreover, we observed no evidence of abnormal NREM sleep fragmentation in any stimulation group over time (RM-ANOVA, *time **

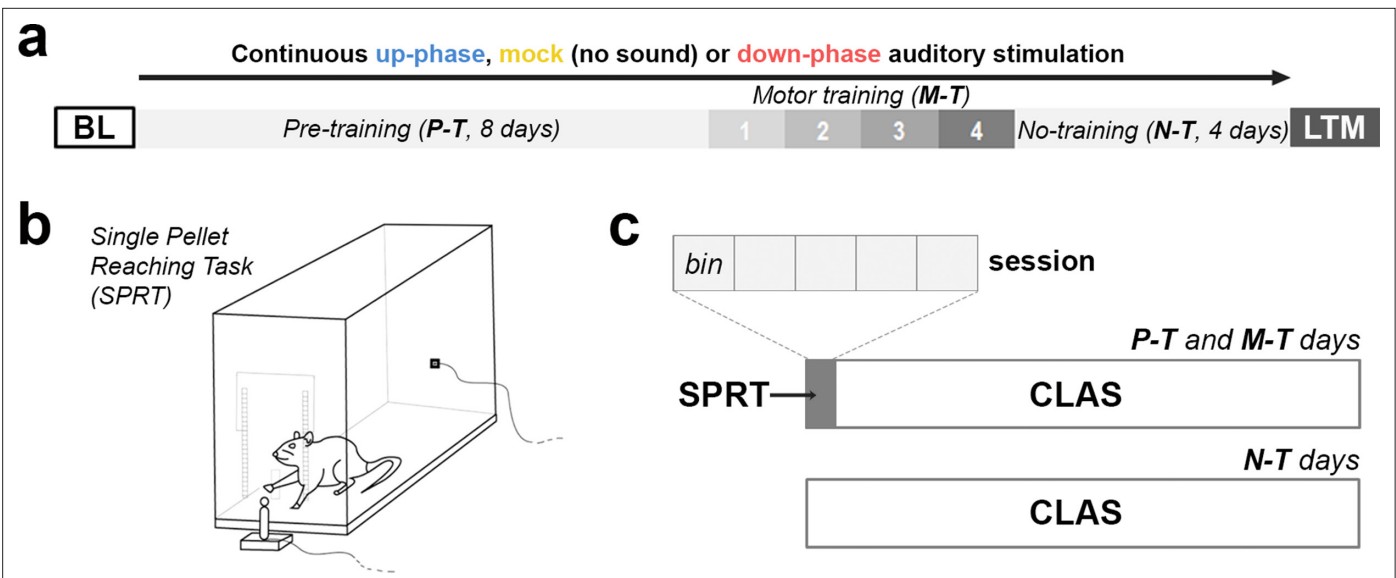

**Figure 2.** Behavioral experimental design. (a) During the course of 16 days under continuous auditory stimulation, the animals were trained in the single-pellet reaching task (SPRT) for the initial 12 days (8 days of pre-training [P-T] and 4 days of motor-training [M-T]). For the remaining and final 4 days, the non-training (N-T) phase, the animals did not perform any behavior test, although they were still kept under auditory stimulation. The protocol was terminated on the 16th day, with one last motor assessment (long-term memory [LTM]). (b, c) From protocol days 1–12, the animals were trained in the SPRT during the first hour of the light period and got back to the auditory stimulation chambers for the remaining time. Each training session had a cutoff at 60 min, divided into five bins of up to 12 min. BL: baseline.

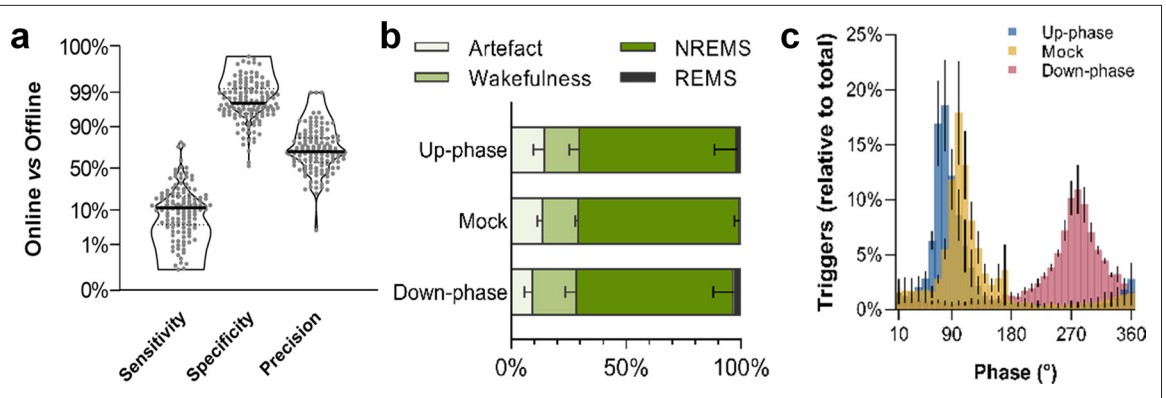

**Figure 3.** Non-rapid eye movement (NREM) sleep staging performance and phase-target distribution. (**a**) Closed-loop auditory stimulation (CLAS) exhibited a 12% sensitivity, 98% specificity, and 70% precision for unsupervised real-time NREM staging (points represent all days from all animals). (**b**) Offline classification of online-labeled NREM sleep epochs per paradigm. . (**c**) Phase distribution at trigger onset for up-phase (targeting 65°), mock (targeting 90° but no delivery of sound), and down-phase (targeting 270°), having defined slow wave's 0° as the rising zero-crossing, 90° to the positive peak, and 270° to the slow-wave trough. For the definitions of sensitivity, specificity, and precision, as well as information about offline scoring of vigilance states and artifact detection, see detailed methods 'EEG/EMG offline scoring and online staging validation' section.

The online version of this article includes the following figure supplement(s) for figure 3:

**Figure supplement 1.** Occurrence and proportion of closed-loop auditory stimulation (CLAS) triggers in wakefulness, non-rapid eye movement (NREM), and rapid eye movement (REM) sleep.

---

*condition* interaction, $F(32, 228) = 0.830$, p=0.730, *Figure 4b*). On third day of motor training (M-T$_3$), 24 hr EEG power spectra in NREM exhibited increased delta activity in the up-phase-stimulated group (*Figure 4c*), whereas down-phase-stimulated animals showed a trend towards decreased delta-frequency activity (*Figure 4e*). Power spectral density in mock animals did not present distinctive changes on day M-T$_3$ (*Figure 4d*). Notably, the stimulation design was accompanied by large inter-individual variability with respect to EEG power in NREM (*Table 1*). To highlight the global effect of each modulatory paradigm, we pooled the daily results for the various sleep-frequency bands, from P-T$_1$ to M-T$_3$ days, and analyzed the average change from BL. Across the spectrum, delta (***p<0.001), sigma (11–16 Hz, ***p<0.001), and beta (16–30 Hz, *p=0.045) bands were significantly increased in up-phase-stimulated subjects. On the contrary, down-phase stimulation decreased delta (***p<0.001), alpha (8–11 Hz, *p=0.037), and sigma (*p=0.012) power during P-T$_1$ – M-T$_3$ days, collectively. Mock animals showed altered power in the sigma band (**p=0.002, one-sample Hotelling's $T^2$, *Figure 4f*), an apparent effect of the skill learning task performed daily. This effect was corroborated by examining sigma activity during the N-T period — 4 days without motor skill training — in all three groups: mock animals returned to BL indices (p=0.672), whereas up-phase (**p=0.002) and down-phase (*p=0.028)-stimulated groups showed increased and decreased sigma activity, respectively (one-sample t-test, *Figure 4g*). Taking into account that auditory stimulation was delivered phase-locked to the EEG's left derivation, we further investigated the possibility of interhemispheric asymmetries upon CLAS. Average power spectral density from stimulated animals was normalized to that of mock to account for effects associated with procedural or instrumental factors unrelated to CLAS, and subsequently compared to a null change (change in percentage points, multiple unpaired t-test followed by Holm–Sidak corrections). Evaluation of protocol day M-T$_3$ showed that both up- and down-phase-stimulated animals presented relatively symmetrical spectral power during wakefulness, NREM sleep, and REM sleep (Figure 4—figure supplement 1a–f). Additionally, we evaluated the impact of behavioral laterality in EEG power, specifically the delta and sigma bands that were found most altered upon CLAS. After splitting the animals within each CLAS paradigm into right- or left-pawed, we considered four scenarios for combined effects of CLAS and handedness (*Vyazovskiy and Tobler, 2008*) in regards to modulation of EEG activity from the left in relation to the right hemisphere (*Figure 4—figure supplement 2a*). Under up-phase CLAS, delta activity was higher on the left EEG derivation in right-pawed animals (case i) than left-pawed ones (case ii), presumably reinforced by the added effect of handedness on the left hemisphere, contralateral to the preferred paw (*Figure 4—figure supplement 2b*). Consistently, down-phase-stimulated animals showed slightly less reduced delta power in the left

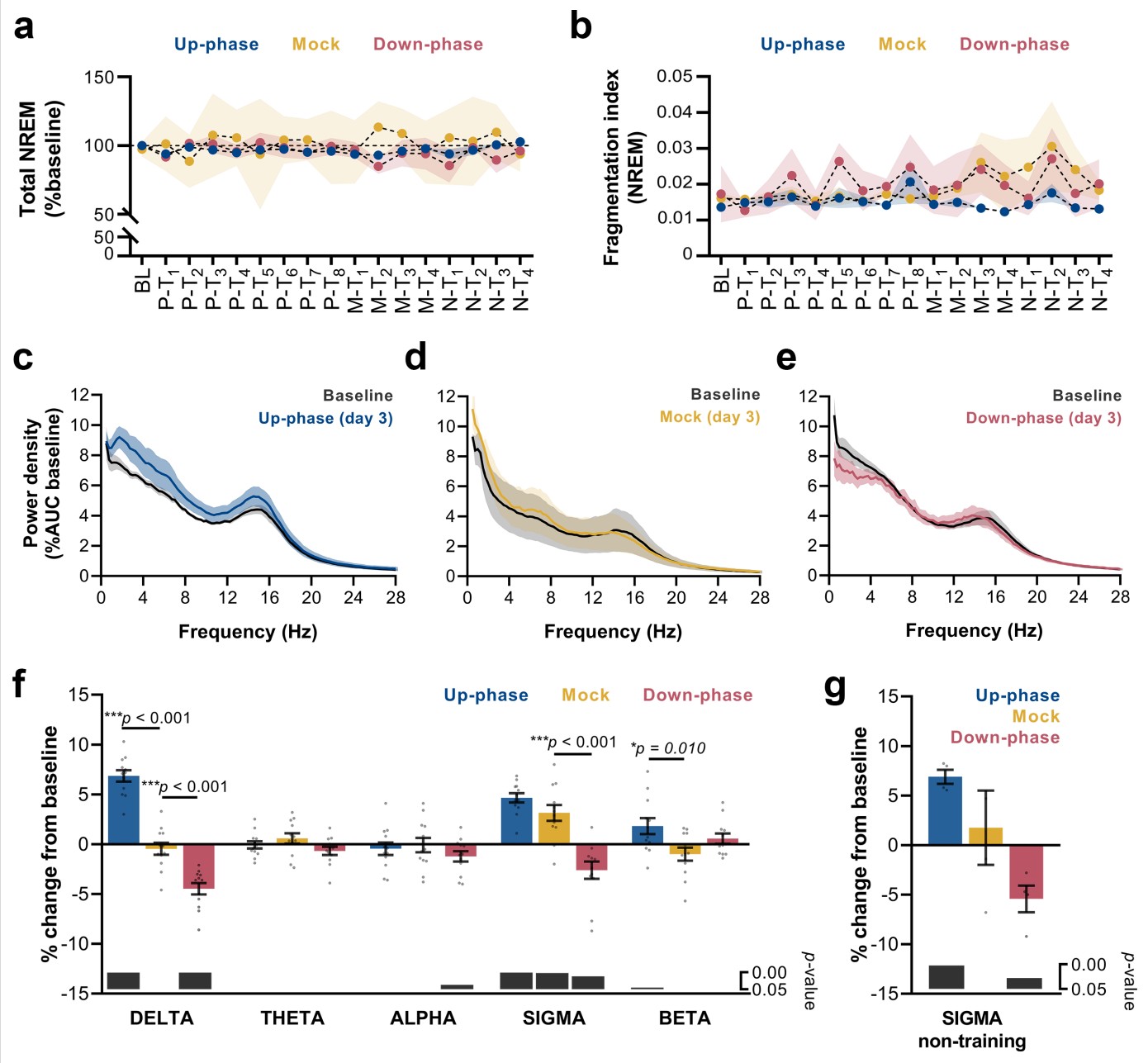

**Figure 4.** Altered delta and sigma power in stimulated animals. (**a**) Non-rapid eye movement (NREM) amount is not different between groups across all protocol days (two-way repeated measures analysis of variance [RM-ANOVA], *condition * day* interaction, $F(32, 215) = 0.929$, p=0.582), (**b**) Fragmentation index reveals no fragmented NREM in none of the groups overtime (RM-ANOVA, *time * condition* interaction, $F(32, 228) = 0.830$, p=0.730). (**c–e**) Average power spectral density in NREM (0.5–28 Hz) extracted from the left electroencephalogram (EEG) derivation during M-T$_3$, relative to baseline (BL). (**f**) 24 hr spectral for all the groups as percentage change from BL for delta (>0.5–4 Hz), theta (>4–8 Hz), alpha (>8–12 Hz), sigma (>11–16 Hz), and beta (>16–20 Hz) frequency bands for all pre-training (P-T) and motor-training (M-T) days (one-sample Hotelling's T$^2$). The change in delta and beta was significantly increased during up-phase stimulation compared to mock (\*\*\*p<0.001 and \*p=0.010), whereas down-phase stimulation showed a significant reduction in relation to non-stimulated animals both in delta and sigma (\*\*\*p<0.001 and \*\*\*p<0.001, respectively). Significant changes from BL are marked as gray bars on the lower part of the chart. (**g**) Sigma power in the mock group back to BL values during non-training (N-T) days (one-sample t-test, p=0.672). AUC: area under the curve.

The online version of this article includes the following figure supplement(s) for figure 4:

**Figure supplement 1.** Interhemispheric symmetry during protocol day M-T$_3$.

**Figure supplement 2.** Exploratory analysis on the effect of behavioral laterality during the single-pellet reaching task on delta and sigma activity from left electroencephalogram (EEG) derivation during M-T$_3$.

**Table 1.** Interindividual and longitudinal variability for delta activity (% change from baseline [BL]). Differences between subjects contribute the most to the variability seen during closed-loop auditory stimulation (CLAS). Values across animals represent the grand average (P-T$_1$ – M-T$_3$) of delta power change from BL per animal. Values across days include a grand average of all animals per day. P-T: pre-training; M-T: motor-training.

| | Across animals (intervariability) | | | | | Across days (P-T1 – M-T33 = 11 days) (longitudinal variability) | | | | |
| --- | --- | --- | --- | --- | --- | --- | --- | --- | --- | --- |
| | Range | Mean | SD | F(DFn, DFd) | p-Value | Range | Mean | SD | F(DFn, DFd) | p-Value |
| Up (n = 7) | +1.11 to +10.45 | +7.36 | 3.19 | (6,61) = 5.26 | <0.001 | +3.04 to +10.3 | +6.85 | 1.98 | (10,57) = 1.66 | 0.114 |
| Mock (n = 5) | −5.32 to +10.99 | +0.14 | 5.91 | (4,40) = 3.51 | 0.015 | −4.60 to +3.32 | −0.46 | 2.11 | (10,34) = 1.22 | 0.313 |
| Down (n = 8) | −8.93 to +1.59 | +4.00 | 3.31 | (7,68) = 4.86 | <0.001 | −8.61 to −2.06 | −4.47 | 1.96 | (10,65) = 0.45 | 0.916 |

hemisphere of right-pawed animals (case iii) than subjects preferring the left paw (case iv) (*Figure 4— figure supplement 2b*). Similarly, sigma power in the left hemisphere was found increased in relation to the right hemisphere in all dextral animals (*Figure 4—figure supplement 2c*).

## Daily time course of delta power changes depends on CLAS' targeted phase, and its 24 hr pattern remains stable across prolonged and continuous stimulation

To understand the daily time course of power density in the delta-frequency band and to appreciate the effect of chronic stimulation during NREM throughout all protocol days, we examined the hourly change in relation to BL for each protocol day. Up-phase-stimulated animals consistently presented increased delta power throughout the day, with larger increases during the light phase (up to ~23% in some days), an effect that persisted until the end of the protocol (*Figure 5a*). Mock animals oscillated around zero-change value (*Figure 5b*), in concordance with the notion of a reorganized sleep-wake pattern, due to the 1-hr-long SPRT task and food-entrainment activity (*Northeast et al., 2019*), the later notably present in the other groups as well. Down-phase-stimulated animals exhibited reduced delta power for the majority of the time, with greater hourly decreases during the dark period, down to ~−17% (*Figure 5c*). We evaluated the time course of delta power from P-T$_8$ to M-T$_{1-4}$ days to explore the effect of stimulation before and after the SPRT hour. In up-phase-stimulated animals, the time-course analysis of delta activity revealed significantly higher power during the hours immediately after the testing session compared to BL (multiple paired t-tests, corrected with Holm–Sidak method for multiple comparisons; *Figure 5d*). This finding overlapped with the animals' largest daily drop in homeostatic sleep pressure (see 'BL curve' section, *Figure 5d*). While mock animals did not show differences during the aforementioned stimulation days compared to BL (*Figure 5e*), down-phase-stimulated animals presented the largest decrease compared to BL during the hours immediately before the SPRT (multiple paired t-tests, corrected with Holm–Sidak method for multiple comparisons; *Figure 5f*). Contrast between delta activity during the SPRT session's preceding and succeeding hours for up- and down-phase stimulation groups shows significant modulation levels for down-phase-stimulated animals in the end of the dark period and for up-phase-stimulated animals immediately after the testing session during the light period (*Figure 5g*).

## CLAS modulates sigma power in a targeted phase-dependent manner

CLAS targeting slow waves also altered sigma power (*Figure 4g*). The hourly time course of sigma-power changes showed a steady boost throughout the experiment, up to ~12% (*Figure 5—figure supplement 1a*). Mock animals exhibited non-significant power changes (range: ~−10% to ~14%) across all stimulation days (*Figure 5—figure supplement 1b*). The decrease in sigma power in the down-phase-stimulated animals reached depressions down to ~−12% during the light period, immediately after the test (*Figure 5—figure supplement 1c*). When we investigated the combined effect of CLAS on P-T$_8$ and M-T$_{1-4}$ days immediately after the training sessions, we found no significant change

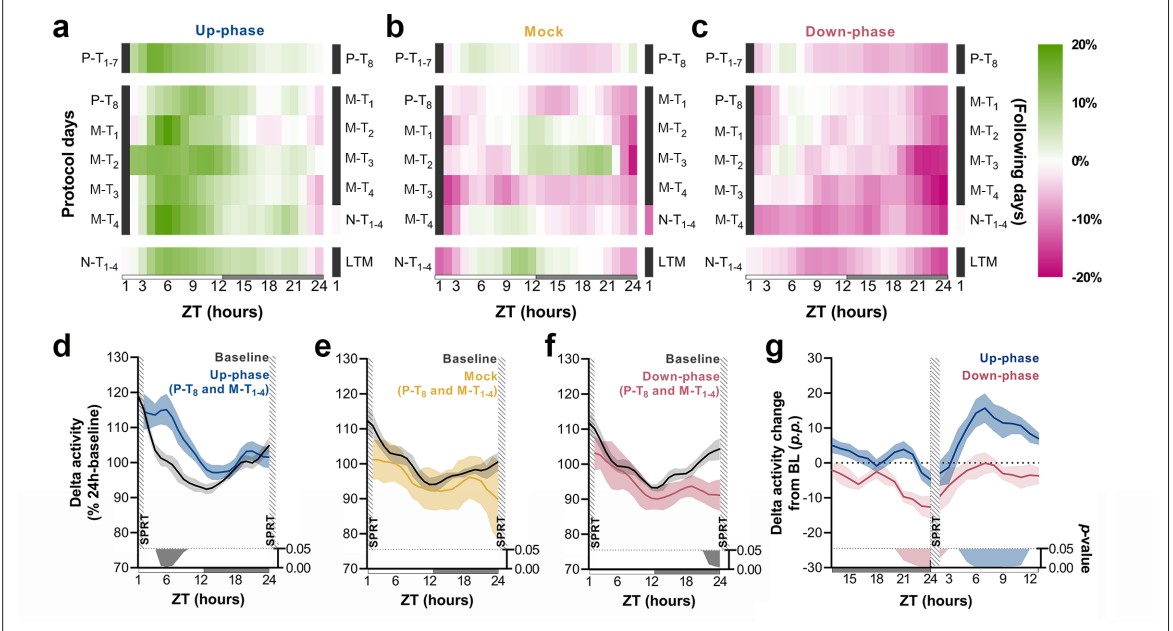

**Figure 5.** Time course of delta activity changes from baseline (BL) during non-rapid eye movement (NREM) sleep. (**a–c**) Delta activity (0.5–4 Hz) changes from BL, in 1 hr bins, for up-phase, mock, and down-phase-stimulated animals. Pre-training (P-T) and non-training (N-T) days were condensed into grand averages (top and bottom heatmap rows, respectively). Individual motor-training (M-T) days are shown in the middle rows. Hours in black represent the single-pellet reaching task (SPRT) training sessions, numbered with the respective protocol day. On the right side of each heatmap, the training sessions of the following protocol day are enumerated. (**d–f**) Time course of hourly delta activity during BL and M-T days, collectively (colored line represents the grand average of training sessions $P-T_8$ to $M-T_4$). These 5 days, as seen individually in the five-row subpanels in the middle of each heatmap (**a–c**), were chosen because they circumscribe the four M-T testing sessions and serve to illustrate the potential effect of modulated NREM sleep on behavior and vice versa. (**g**) Contrast between delta activity during the SPRT session's preceding and succeeding hours for up- and down-phase stimulation groups, plotted in percentage points from BL. Gray/colored shadows bellow plots (**d–g**) represent significant timepoints (multiple t-tests, Holm–Sidak correction for multiple comparisons). The light period comprises ZT-1 to ZT-12 (white rectangle), whereas the dark period spans from ZT-13 to ZT-24 (gray rectangle). ZT: zeitgeber; LTM: long-term memory; p.p.: percentage points.

The online version of this article includes the following figure supplement(s) for figure 5:

**Figure supplement 1.** Time course of sigma activity changes from baseline (BL) during all protocol days and motor-training (M-T) grand average.

in sigma power in the up-phase-stimulated and mock groups compared to their BL (multiple paired t-tests, Holm–Sidak method for multiple comparisons, *Figure 5—figure supplement 1d and e*). On the contrary, down-phase stimulation reduced sigma power compared to BL, particularly during the hours preceding the motor-training task (multiple paired t-tests, Holm–Sidak method for multiple comparisons, *Figure 5—figure supplement 1* and *Figure 5—figure supplement 1g*).

## Up-phase CLAS preserves the natural proportion of same-size trains of triggers, whereas down-phase CLAS prolongs the interval between triggers

Next, we examined the frequency of trains of triggers (interstimulus interval [ISI] ≥ 0.8 s): mock animals show a skewed distribution, with single triggers representing ~65% of all sequences of triggers and ~43% of all triggers in NREM ($M = 2384$, SD = 2296, per 24 hr). Up-phase-stimulated animals showed no difference in proportion of trains of any size when compared to the mock group (*Figure 6a*); however, they overall received more triggers per day ($M = 5862$, SD = 2214). Conversely, down-phase-stimulated subjects presented a different pattern of stimulation (two-way ANOVA, *condition * train size* interaction, $F(8, 80) = 6.72$, ***p<0.001, *Figure 6a*), with significant higher rate of single triggers (***p<0.001, ~58% of all triggers in NREM per 24 hr, $M = 2462$, SD = 1686) and less trains of two triggers (*p=0.020), hinting on the disruptive effects of down-phase targeting on slow waves. Corroborating these results, the analysis of ISIs (two-way ANOVA, *condition * ISI* interaction, $F(16, 144) = 1.80$, *p=0.037) showed that up-phase targeting preserved the proportion of

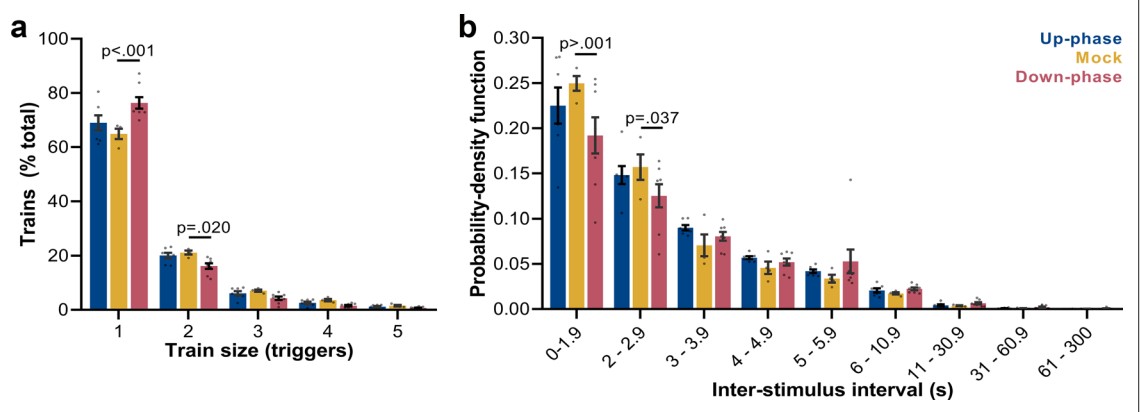

**Figure 6.** Trigger's entrainment and distribution. (**a**) Histogram for the train of pulses, up to five consecutive stimuli, spaced by no more than 1 s (two-way ANOVA, *condition * train size* interaction, $F_{(8, 80)} = 6.72$, ***p<0.001). (**b**) Histogram for the time between triggers, organized in a probability density function for different bin sizes (two-way ANOVA, *condition * interstimulus interval* interaction, $F_{(16, 144)} = 1.80$, *p=0.037).

The online version of this article includes the following figure supplement(s) for figure 6:

**Figure supplement 1.** Grand mean event-related potential (ERP) of up- and down-phase groups, and amplitude contrast between the two conditions.

same-length non-stimulated periods seen in the analysis of muted-sound triggers in non-stimulated subjects, whereas down-phase stimulation provided a distribution of triggers containing significantly less density of ISIs < 2 s (**Figure 6b**).

## Up-phase CLAS mildly entrains EEG slow waves, whereas stimulation during the down-phase disturbs it

Analysis of event-related potentials (ERPs, filtered in the 0.5–2 Hz frequency band), that is, all waveforms receiving triggers during NREM, revealed a small slow-wave cycle following the endogenous cycle that triggered the stimulation in up-phase CLAS (**Figure 6—figure supplement 1a**), while down-phase targeting appeared to completely disrupt the slow-wave negative half-cycle (**Figure 6—figure supplement 1b**). This is evidenced by the amplitude contrast between the two conditions for two succeeding slow-wave troughs (**Figure 6—figure supplement 1c**). Although not statistically

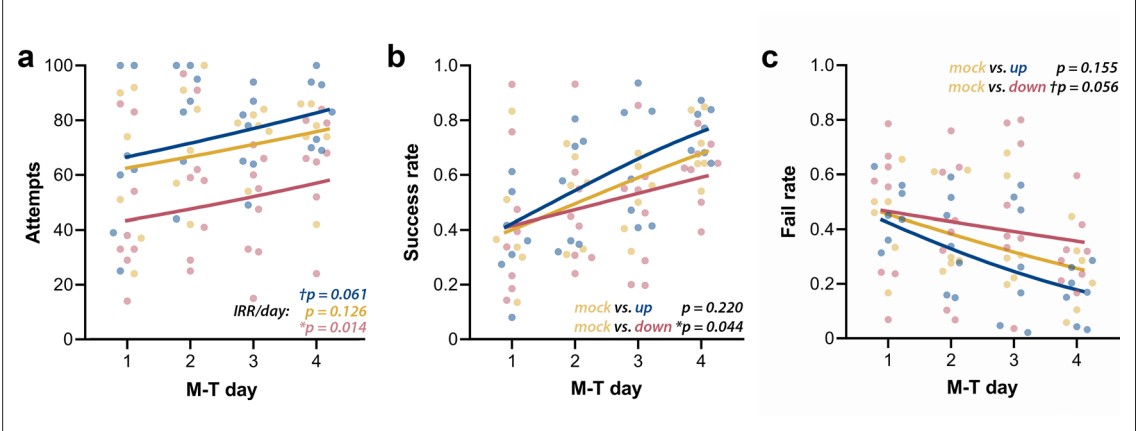

**Figure 7.** Up-phase-stimulated rats' learning rate progresses similarly to controls over time, whereas that one of down-phase-stimulated rats is impaired. (**a**) The number of attempts in up-phase (n = 7, $†p_{IRR/day} = 0.061$) and mock (n = 6, $p_{IRR/day} = 0.126$)-stimulated rats did not significantly change over the period of the test, whereas down-phase rats showed a significant increase in this parameter (n = 8; $*p_{IRR/day} = 0.014$). (**b**) By observing success rate (SR) (number of successes in total attempts), down-phase-stimulated subjects show significantly worse learning progression when contrasted with mock subjects (($X^2(2, N = 21) = 17.50$, ***p<0.001; Tukey post-hoc comparison *up * mock*: p=0.220, *down * mock*: *p=0.044). (**c**) Like SR results, the fail rate (FR) analysis ($X^2(2, N = 21) = 18.34$, ***p<0.001; Tukey post-hoc comparison *up * mock*: p=0.155, *down * mock*: †p=0.056) shows impaired learning in the down-phase-stimulated group compared to mock, while up-phase performs similarly to controls. M-T: motor-training; IRR/day: incidence rate ratios per day.

significant, we observed trends for lower amplitude for both the depolarized up-state (unpaired t-test, p=0.065) and the hyperpolarized down-state (unpaired t-test, p=0.058) in down-phase-stimulated animals compared to up-phase-stimulated rats.

## Phase-targeted CLAS modulates success rate of motor learning in the SPRT

We first analyzed the raw counts for attempts, successes, failures, and drop-ins. Attempts' counts had high dispersion for every M-T day, and although they showed considerable qualitative clustering at day 4 (*Figure 7a*), their high dispersion did not allow differentiation of the groups. The lack of apparent interaction between *group* and *day* was confirmed by the negative binomial distribution model ($X^2$(2, N = 21) = 0.25, p=0.883). Increase in the incidence rate ratios (IRRs) of attempts across M-T days (IRR/day) did not achieve significance for mock animals (n = 6; $p_{IRR/day}$ = 0.126), but revealed a tendency of growth in the up-phase-stimulated group (n = 7; $\dagger p_{IRR/day}$ = 0.061) and a significant increase in the down-phase-stimulated group (n = 8; $*p_{IRR/day}$ = 0.014). Successes' count grew in all groups ($X^2$(1, N = 21) = 50.08, ***p<0.001), but without significant interaction ($X^2$(2, N = 21) = 0.55, p=0.761), rendering in neither up- (p=0.613) or down-phase (p=0.123)-stimulated groups significant differences from mock animals. Failed attempts' counts showed differences between groups across training days ($X^2$(2, N = 21) = 8.29, *p=0.016). This result reflects the decrease in the incidence of fails in the up-phase-stimulated group (***$p_{IRR/day}$ < 0.001), but not in the down-phase group ($p_{IRR/day}$ = 0.864), with mock intermediate to both (n = 6; $\dagger p_{IRR/day}$ = 0.054). Drop-in counts (the pellet is grasped but dropped during arm retraction) present only a collective reduction over time ($X^2$(1, N = 21) = 4.42, *p=0.035), without differences between groups. Between-group comparison of success rate (SR, successes per total attempts) and fail rate (FR, fails per total attempts) demonstrated that down-phase-stimulated subjects present a significantly worse learning progression when contrasted with mock subjects (SR: $X^2$(2, N = 21) = 17.50), ***p<0.001; Tukey post-hoc comparison *up * mock*: p=0.220, *down * mock*: *p=0.044, *Figure 7b*; FR: $X^2$(2, N = 21) = 18.34, ***p<0.001; Tukey post-hoc comparison *up * mock*: p=0.155, *down * mock*: $\dagger$p=0.056, *Figure 7c*. SR (two-way ANOVA, *time * condition*, $F$(2, 18) = 0.14, p=0.874) and FR (two-way ANOVA, *time * condition*, $F$(2, 18) = 0.08, p=0.920) at LTM-day did not significantly differ from M-T$_4$ in any of the groups.

## Up-phase-stimulated animals show daily intra-session improvements, while down-phase-stimulated animals attempt less towards the end of the session

Compartmentalizing each training session into 12 min bins (or 20 pellets, whichever comes first) revealed that up-phase-stimulated and mock subjects maintained the same rate of attempted grasps across bins (RM-ANOVA, *condition * bin* interaction, $F$(2, 18) = 4.03, *p=0.036, *Figure 8a*), while down-phase-stimulated animals attempted significantly less towards the end of the session (*first-bin * last-bin*, *p=0.022) and significantly less than control animals during the last bin (*down * mock*, *p=0.037). When relativizing number of attempts per bin by the total number of attempts within the session (RM-ANOVA, *condition * bin* interaction, $F$(2, 18) = 5.24, *p=0.016, *Figure 8b*), down-phase-stimulated animals again show a sharp decrease towards the end of the session (*first-bin * last-bin*, **p=0.004). Attempting speed (number of attempts per minute) also showed a decline from the first to the fifth bin in the down-phase-stimulated subjects, but failed to reach significance (RM-ANOVA, *condition * bin* interaction, $F$(2, 18) = 1.33, p=0.291, *Figure 8c*). Although the intra-session attempt rate in the up-phase-stimulated group remained constant from the beginning to end of training sessions, there was an increase in successful attempts from the first to the fifth bin in this group in all M-T days. Conversely, down-phase-stimulated animals were always slightly less successful towards the end of each session (*Figure 8d*), although this observation might be explained by the reduction in attempts along the session duration.

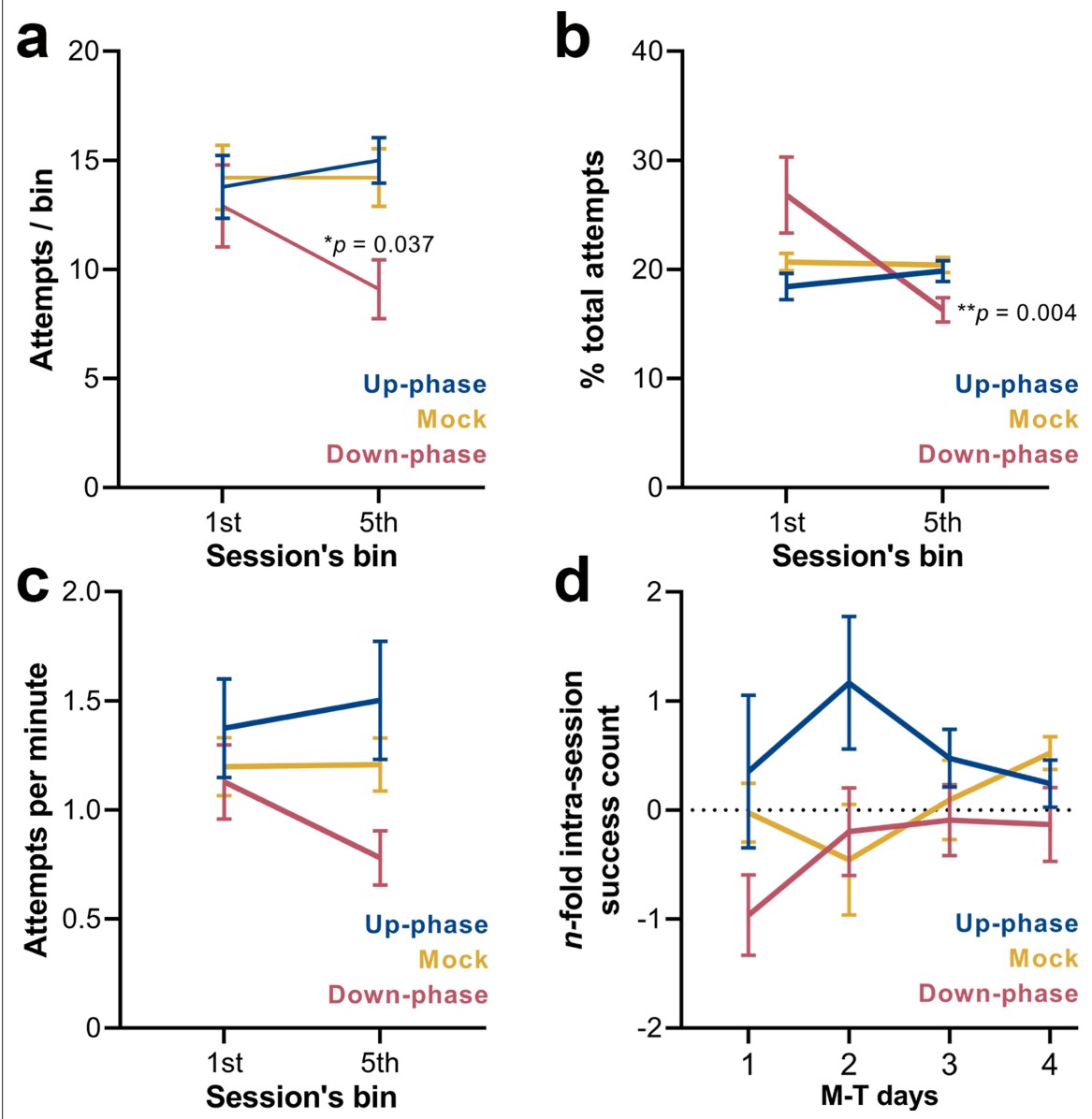

**Figure 8.** Down-phase-stimulated animals show a reduction in the engagement with the task towards the end of the session. (**a**) Down-phase-stimulated animals attempt significantly less than control animals during the last bin (two-way repeated measures analysis of variance [RM-ANOVA], *condition * bin* interaction, $F_{(2, 18)} = 4.03$, *p=0.036; *down * mock*, *p=0.037). (**b**) In relation to the proportion of attempts per bin to the total number of attempts, down-phase-stimulated animals again show a sharp decrease towards the end of the session (RM-ANOVA, *condition * bin* interaction, $F_{(2, 18)} = 5.24$, *p=0.016; *first-bin * last-bin*, **p=0.004). (**c**) Attempting speed shows a non-significant decline from the first to the fifth bin in the down-phase-stimulated subjects (RM-ANOVA, *condition * bin* interaction, $F_{(2, 18)} = 1.33$, p=0.291). (**d**) Up-phase-stimulated animals show a consistent increase in successful attempts from the first to the fifth bin, in all motor-training (M-T) days. Conversely, down-phase-stimulated animals were always slightly less successful towards the end of each session, although this might be due to the drop in attempts within the last testing window (plotted as the $\log_2$ of the fold-change of the number of successes of the last bin to the first bin of a session: –1 represents a 50% drop in successes, while 1 represents twice more successes from first bin).

## Number of triggers correlates with magnitude of delta power changes, and both delta and sigma changes positively correlate with success rate at the end of the M-T period

To investigate whether the number of sound triggers influences the extent of delta power modulation, we extracted the number of triggers over each 24 hr period during the M-T phase for all stimulated animals. The magnitude of changes in the delta-frequency band was moderately correlated with the

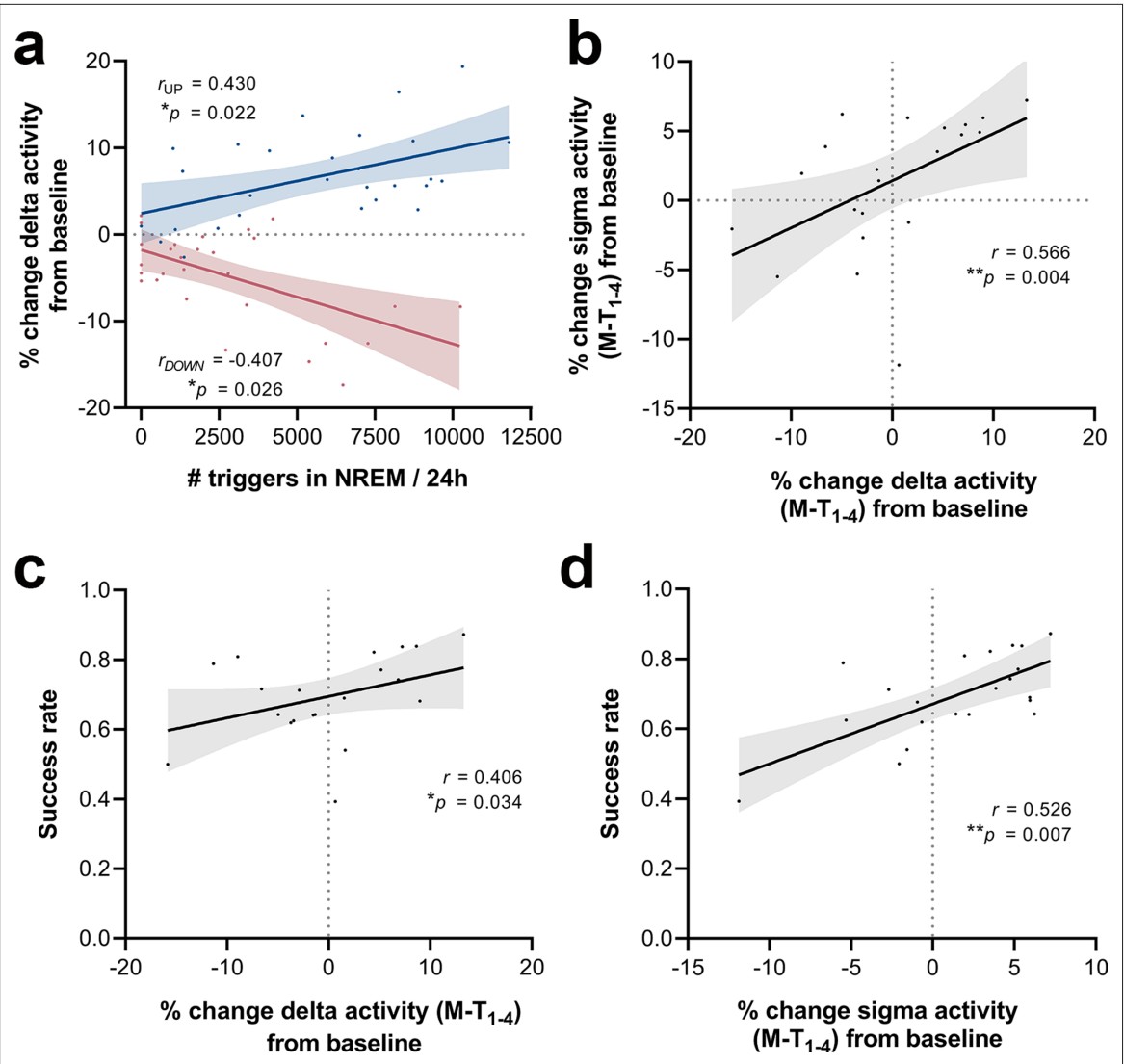

**Figure 9.** Correlations between number of triggers, delta activity change from baseline (BL), sigma activity change from BL and success rate. (**a**) The magnitude of changes in the delta-frequency band was moderately correlated with the number of stimuli delivered, both for up- (Spearman's rho = 0.43, *p=0.022) and down-phase-stimulated animals (Spearman's rho = −0.41, *p=0.026). (**b**) In turn, 24 hr grand average (motor-training [M-T] days) of delta activity changes from BL correlated significantly with sigma activity changes in the same period (Spearman's rho = 0.57, **p=0.004). (**c, d**) Moderate positive associations between the success rate (SR) scores achieved by all subjects in the last motor assessment (M-T$_4$) and the global changes in delta activity (Spearman's *rho* = 0.41, *p=0.034) and sigma activity (Spearman's *rho* = 0.53, *p=0.007), from all days preceding the sessions of the M-T phase, collectively (P-T$_8$, M-T$_1$, M-T$_2$, and M-T$_3$).

number of delivered stimuli, both for up- (Spearman's rho = 0.43, *p=0.022, **Figure 9a**) and down-phase-stimulated animals (Spearman's rho = −0.41, *p=0.026, **Figure 9a**). We also found a significant positive association between 24 hr changes in delta and sigma power (Spearman's rho = 0.57, **p=0.004, **Figure 9b**), reflecting the impact of CLAS on the two most dominant features during NREM sleep. Subsequent analysis between 24 hr average delta change (all training-phase days condensed) and SR at the end of the training phase revealed a moderate positive association between modulation of delta activity and SPRT performance (Spearman's rho = 0.41, *p=0.034, **Figure 9c**). Likewise, sigma activity changes positively correlate with SR scores (Spearman's rho = 0.53, *p=0.007, **Figure 9d**).

## Discussion

### Conceptual and technical validation of rodent CLAS: chronic continuous phase-targeted CLAS specifically and steadily modulated delta and sigma power

This is the first report of successful phase-targeted CLAS in rodents. With this paradigm, we delivered auditory triggers to healthy rats continuously over a period of 16 days. We observed a mild but specific average increase of 6.9% (up to 23% during the light period) in delta power per 24 hr on stimulation days compared to BL. This effect was stable over the entire stimulation period evaluated, with no signs of decay nor facilitation effects over time. Reverse effects ($M = -4.5\%$, down to $-17.5\%$ by the end of the dark period) were observed during down-phase stimulation.

Notably, application of CLAS in human subjects has often failed to demonstrate an overall inter-night effect (unperturbed night vs. stimulated night) of stimulation on power spectra (*Papalambros et al., 2017*; *Papalambros et al., 2019*), while comparison between same-night ON- and OFF-stimulation windows yielded significant stimulation effects that positively correlated with subsequent memory performance (*Ong et al., 2016*; *Papalambros et al., 2017*). In this study, we provide first evidence of a long-term effect when comparing unperturbed 24 hr (BL) vs. stimulated 24 hr in a paradigm of continuous stimulation, somewhat challenging the proposition that the effect of CLAS is mediated by qualitative transient reorganization of SWA rather than of a general SWS or SWA enhancement (*Papalambros et al., 2017*). CLAS elicited, however, a modest effect in both up- and down-conditions in our rats, without evidence of either build-up or adaptation mechanisms to regulate the impact of the stimulation across the days. Interestingly, chronic sleep deprivation studies in rodents demonstrate a comparable effect, suggesting that strong homeostatic compensatory mechanisms may be at play (*Kim et al., 2020*; *Leemburg et al., 2010*).

To achieve a greater effect, further refined CLAS paradigms must be evaluated. For instance, although the CLAS protocol employed in this study accomplished 70% NREM precision for sound triggering and clear phase-target accuracy, future NREM online staging and CLAS pipelines shall be optimized in terms of detection of particular EEG events, such as slow oscillations, spindles, or arousals. Triggers would ideally be delivered only during sustained NREM sleep, but motion behaviors with a rhythmic nature (e.g., grooming, feeding, and drinking) may have been major factors contributing to wakefulness and artifact epochs being identified online as NREM sleep, and consequently reducing precision. Thus, further artifact-rejection features shall be considered, such as auxiliary EMG information (e.g., piezo-electric sensors), especially in long-term experiments where attrition of implanted EEG/EMG headsets is unavoidable. In addition, very strict thresholds for NREM sleep online identification drastically limited sensitivity of the algorithm (12%). Nonetheless, the CLAS paradigm introduced here produces stable sleep modulation effects over time and successfully replicates the timing accuracy so far obtained in some methods of human CLAS (*Fattinger et al., 2019*; *Papalambros et al., 2017*). Also like in humans studies (*Henin et al., 2019*; *Krugliakova et al., 2020*; *Papalambros et al., 2017*), we observed an additional modulatory effect of CLAS on power in the sigma-frequency band in rats. Sigma modulation was, just like CLAS-dependent delta changes, phase-target-specific and steady over the 16 days of stimulation. Finally, further assessments indicated that exclusively targeting slow waves in the left hemisphere did not manipulate SWA during NREM sleep to a different extent in that brain area. However, a left predominance of SWA—presumed to correspond with the naturally higher occurrence of right-pawed animals—has been reported in rodents (*Vyazovskiy et al., 2002*; *Vyazovskiy and Tobler, 2008*), and therefore we cannot rule out the possibility that this natural characteristic mounts up with CLAS effect and entangles with our results.

We explored the size of trains of triggers and ISIs (*Santostasi et al., 2016*)—proxies for length and distribution of delta wave trains—to explore whether CLAS altered natural slow-wave patterns and if this could be the potential underlying process by which CLAS exerted its modulatory effect over delta power. In comparison to mock, down-phase-stimulated animals showed an increased number of single triggers, decreased proportion of trains of two triggers, and decreased density of <2 s ISIs, in contrast to maintained levels in the up-phase-stimulated group, potentially indicating that down-phase CLAS has disarrayed the natural pattern of delta wave trains.

In agreement with the apparent preservation of the slow-wave pattern with up-phase CLAS, and opposite effect with down-phase CLAS, analysis of ERPs indicated the occurrence of a post-trigger

slow wave when targeting 60°, whereas a post-trigger pattern disruption arises from 180° target stimulation. Peak-to-peak amplitude did not differ between up- and down-phase groups, although we cannot reject a possible tapering of the physiological response from stimuli outside of the desired phase. Taken together, these findings suggest that CLAS of rodent slow waves may be altering the natural slow waves' pattern rather than significantly affecting their amplitude.

## Functional validation rodent CLAS: altered behavioral performance upon phase-targeted CLAS in healthy rats

Phase-targeted CLAS produced significantly altered learning performance in the SPRT in healthy rats, mainly in the down-phase condition. Moreover, we found that the mean delta activity change in motor-training days 1–4 in relation to BL correlated positively with SR on motor-training day 4 in the SPRT. Through the positive correlation found between the number of triggers—that is, extent of closed-loop auditory stimuli delivered—and delta power changes in up- and down-phase groups, we confirmed indeed that delta activity changes during motor-training days 1–4 are actually driven by CLAS. Overall, these results suggest that CLAS-dependent delta activity levels drive next-day performance in healthy rats. Furthermore, we observed that the level of proficiency achieved at motor-training day 4 was maintained in all groups after 5 days without training. This observation suggests that CLAS modulated sleep-dependent skill acquisition and retention only immediately following a motor-training session. In short, without prior active task engagement or performance saturation (*Kvint et al., 2011*), up-phase CLAS per se seems insufficient to induce a significant performance enhancement at the present conditions, whereas there was no evidence of skill decay induced by down-phase CLAS either. Two sleep-dependent mechanisms may explain the observed effect of modulated slow waves on behavioral performance in our paradigm. Sleep-mediated synaptic downscaling, or recovery, after learning has been deemed critical to maintaining behavioral performance (*Fattinger et al., 2017*). Thus, it is conceivable that targeting slow waves with CLAS might have altered (up-: increased; down-: decreased) the normal recovery process after learning during wakefulness in our rats. In support of this argument, we observed that the number of attempts in the down-phase group seems lower than in the other groups at motor-training day 1, potentially indicating that before any memory consolidation process took place, these animals already performed worse than their counterparts did. On the other hand, memory consolidation, a process at least partially mediated by spindles (*Diekelmann and Born, 2010*) in which specific synapses are strengthened, is also potentially sensitive to CLAS (*Krugliakova et al., 2020*; *Ngo et al., 2013a*; *Papalambros et al., 2017*). In fact, we did observe positive correlations between sigma—potentially a proxy for spindles (*Holz et al., 2012*)—during motor-training days 1–4 and SR on motor-training day 4. The two theories, however, are certainly not mutually exclusive (*Klinzing et al., 2019*; *Wilckens et al., 2018*) and sleep-dependent memory benefit could well complementarily rely on them both. Our tool, thus, may be utilized to disentangle the exact effects of CLAS-modulated SWS on rodent cognition. Further, CLAS can be strategically combined with other tools, such as multiunit-activity recording, and provide crucial insight into the underlying neuronal mechanisms of auditory-modulated slow waves and sigma (*Krugliakova et al., 2020*), and their functional role on learning and memory.

### Limitations

An experimental limitation is the fact that we did not actively apply randomization, nor exclusion criteria, in relation to the paw preference in our experimental groups of all-male rats. Thus, as we solely tracked the left-hemisphere EEG derivation for CLAS, distinct local network dynamics determined by paw dexterity (*Hanlon et al., 2009*; *Vyazovskiy and Tobler, 2008*) may have potentially affected closed-loop stimulation features in an uncontrolled manner in left- or right-pawed experimental subjects and contributed synergistically or in opposition to the CLAS modulatory effects. Larger group numbers allowing for exclusion or efficient randomization of this confounder variable shall be considered in future efforts. Lastly, in our temporal analysis of the delta power changes per 24 hr, we observed a clear pattern showing that animals in all groups present distinctively decreased delta power levels in the few hours before the behavioral test. Whether this activity pattern, likely attributable to food anticipation (*Northeast et al., 2019*), might have compounded with our CLAS-specific delta power findings remains to be determined and, ideally, further controlled in future experiments. Finally, in relation to the lack of effect of phase-targeted rodent CLAS on ERP amplitudes, as

others demonstrated earlier, the broad range achieved by our phase-tracking algorithm might have contributed to the observed variable effects of CLAS across animals at the ERP level, ultimately leading to high interindividual variability on delta activity modulation, result that agrees with previous studies using non-adaptive prediction algorithms (*Cox et al., 2014*; *Fattinger et al., 2017*; *Fattinger et al., 2019*; *Ngo et al., 2013a*). Employing adaptive feedback methods based on phase-locked loops (at times associated with amplitude thresholds), producing optimal phase-prediction and inducing more specific and systematic effects on SWS (*Ong et al., 2016*; *Papalambros et al., 2017*; *Papalambros et al., 2019*; *Santostasi et al., 2016*), shall be assessed in rodents next.

Altogether, our report presents the first successful implementation of phase-targeted CLAS in rats, demonstrating that phase-targeted continuous chronic CLAS is feasible, specific, and effective in rodents. Overall, this novel tool provides successful grounds to not only further study the substrate mechanisms by which CLAS affects SWA and associated outcomes, such as cognition, but to additionally assess its effect on models of brains disease. Ultimately, CLAS of slow waves in rodents may help pave the way to the clinical implementation of this novel technique in the context of brain/body disorders that may be targeted with sleep-based therapeutics (*Arora and Taheri, 2015*; *Cook et al., 2020*; *Fung et al., 2011*).

## Materials and methods

### Animals, surgeries, and husbandry

To develop this tool, we used 23 young adult male Sprague–Dawley rats (Charles River, Italy) weighing 200–220 g and group-housed them in standard IVC cages (T2000) prior to interventions. In all animals, we surgically implanted electrodes for continuous recording of EEG/EMG as described previously (*Büchele et al., 2016*). Briefly, we inserted four stainless steel miniature screws (Hasler, Switzerland), one pair for each hemisphere, into the rats' skull following specific stereotactic coordinates: the anterior electrodes were implanted 3 mm posterior to bregma and 2 mm lateral to the midline, and the posterior electrodes 6 mm posterior to bregma and 2 mm lateral to the midline. For monitoring of muscle tone, we inserted into the rats' neck muscle a pair of gold wires that served as EMG electrodes (*Figure 1a*). All electrodes were connected to stainless steel wires, further connected to a headpiece (Farnell, #M80-8540842, Switzerland) and fixed to the skull with dental cement. We performed all surgical procedures under deep anesthesia by inhalation of isoflurane (4.5% for induction, 2.5% for maintenance), and subsequent analgesia with buprenorphine (s.c., 0.05 mg/kg). Following surgery, we housed the animals individually for a minimum of 14 days for recovery, with food and water available ad libitum, and handled them daily for postoperative monitoring, body weight checkup, and familiarization with the experimenter. The animal room temperature was maintained at 22–23°C, and animals were kept on a 12 hr light-dark cycle. 2 weeks after surgery, we transferred the animals, still individually, to custom-made acrylic-glass cages (26.5 × 42.5 × 43.5 cm). Each cage was positioned inside a sound-attenuated chamber, built in-house, for EEG/EMG recording and auditory stimulation (*Figure 1b*). All procedures were approved by the veterinary office of the Canton Zurich (license ZH231/2015) and conducted in accordance with national and institutional regulations for care and use of laboratory animals.

### Sound insulation chambers

Eight 40 × 50 × 60 cm sound-insulated chambers were design and built in-house with the purpose to simultaneously record physiological signals and deliver auditory stimulation to eight individual subjects at a time. Each chamber (*Figure 1—figure supplement 1a*) was built using the following components:

- PVC boards (1 cm thickness, 0.60 g/cm³; KÖMATEX, Röhm, Switzerland), to form a lightweight but robust structure.
- One layer of absorptive closed-cell foam (2 cm thickness, 0.02 g/cm³, PU WAVE TOP, Swilo, Switzerland), covering the entire inner surface of the chamber, and a soundproofing closed-cell foam with protective skin and heavy rubber layer (3.2 cm thickness, 0.03 g/cm³, PU SKIN, Swilo, Switzerland) on the outer surface for reduction of sound and vibration transmissions.
- One fully demountable door, with the same isolation as described above.

- LED strips (Philips LightStrips Flex Color), attached around the chamber's ceiling, were connected to an intensity controller and dedicated timer for light/dark cycle. The light was evenly diffused using a translucent acrylic plate (3 mm thickness, $T(D_{65})$ = 72%, Plexiglas, Röhm, Switzerland).
- An additional infrared LED was also mounted to allow for continuous infrared video recordings.
- Passive slip-ring (Dragonfly Research, USA) for EEG/EMG recordings in freely moving animals.
- Two low-noise fans (air flow 73.63 m$^3$/hr, SPL = 5.71 dB/A, NB-eLoop 140 mm, Blacknoise, Germany), for air in- and outlet.
- A recording camera (GigE Basler ½" NIR, NOLDUS, Germany), equipped with a varifocal lens (4.5–12.5 mm, NOLDUS) and infrared pass filter.

## Pilot studies

We performed a series of small studies in a cohort of 10 animals (five animals under up-phase and five under down-phase CLAS) to investigate the most adequate set of stimulation parameters. Briefly, we opted to stimulate at 35 dB (range tested: 50 dB, 45 dB, 40 dB, and 35 dB) based on real-time video and posterior EEG/EMG inspection as none of the animals showed any perturbation during sustained NREM sleep under this volume. Based on literature (*Fattinger et al., 2017*; *Ngo et al., 2013a*), and considering average delay from detection to trigger of 29.30 ms, we selected 60° for up-phase and 180° for down-phase stimulation (we defined slow wave's 0° as the rising zero-crossing, 90° to the positive peak, and 270° to the slow-wave trough). Phase-target CLAS in humans typically uses 50 ms stimulus duration for a dominant slow oscillatory component of 0.8 Hz. Therefore, we determined 30 ms of stimulus duration to overlap with the ongoing slow wave proportionally—slow oscillatory activity peaks at 1.35 Hz in rats (*Mölle et al., 2009*).

## Experimental design

We performed the whole experiment in four batches of five or six animals each, divided evenly across experimental groups. Aiming to test several subjects daily, individually and at the same circadian time, the light/dark cycle of each subject was adapted accordingly. Briefly, we shifted the light/dark cycle at once, allocating the animals' first hour of light into successive testing windows of 1.5 hr during the course of the working day. Bearing in mind that, during the testing period, the circadian cycle of the last animal of each batch was shifted 9 hr in relation to the first, the entire batch was given a 10-day adaptation to the new routine, as well as to the recording chamber and cables. We kept the animals on a feeding schedule in which they daily received 50 g/kg of regular chow after training throughout the entire experiment. This feeding schedule did not cause any significant weight loss. Water was available ad libitum. Following 24 hr of BL recording, we initiated a 16-day auditory stimulation protocol in parallel with a fine motor-skill learning task (see 'Single-pellet reaching task protocol' section). On the 17th protocol day, we performed the last behavioral assessment and sacrificed the animals (*Figure 2*).

## EEG/EMG recording and preprocessing

In order to verify the effect of auditory stimulation on EEG spectra, we conducted bilateral tethered EEG/EMG recordings (differential mode) during 24 hr, to serve as BL, and throughout all the subsequent 16 protocol days, applying our runtime stimulation paradigm in five or six freely moving animals simultaneously. We acquired data using a Multichannel Neurophysiology Recording System (Tucker Davis Technologies [ TDT], USA). We sampled all EEG/EMG signal at 610.35 Hz, amplified (PZ5 Neuro-Digitizer preamplifier, TDT) after applying an anti-aliasing low-pass filter (45% of sampling frequency), synchronously digitized (RZ2 BIOAMP processor, TDT), recorded using SYNAPSE software (TDT), and stored locally (WS-8 workstation, TDT). We filtered real-time EEG between 0.1 and 36.0 Hz (second-order biquad filter, TDT), and EMG between 5.0 and 525.0 Hz (second-order biquad filter and 40 dB notch filter centered at 50 Hz, TDT), and feed the signals to real-time detection algorithms for NREM staging and phase detection (*Figure 1c*).

## Online NREM staging

Parallel rule-based NREM staging and phase detection features were continuously running alongside EEG/EMG recording, that is, sound triggers were presented at every instance the stimulatory truth function compounding these features was reached (*Figure 1c*, *Figure 1—figure supplement 1*). For

online sleep staging, a nonlinear classifier compounds two major decision nodes: power in EEG and power in EMG (*Hamrahi et al., 2001*). Briefly, we computed high-beta (20–30 Hz) and delta (0.5–4 Hz) bands' rms on a sliding window of 1 s using an algorithm written in RPvdsEx (Real-Time Processor Visual Design Studio, TDT). Once the ratio $rms_{delta}/rms_{high-beta}$, hereinafter referred to as NREMratio, crossed a threshold individually identified during the BL recording, we further confronted EMG rms to a threshold, as well as defined during BL, in order to rule out movement artefacts. The abovementioned NREMratio and the EMG power thresholds for online NREM staging during auditory stimulation, suggestive of sustained SWS during NREM, were extracted individually and immediately after the BL recording: 24 hr EEG/EMG data was automatically scored (SPINDLE, ETH Zurich, Switzerland) and fed to a custom-written MATLAB (ver. R2016b) script. In short, a strict estimate of NREMratio threshold during NREM was established as +1.0 SD over the mean (representing the 84.1% percentile) of the NREMratio of all NREM epochs during BL. Similarly, EMG power in NREM was delimited to values –1.0 SD below the mean (threshold at 15.9% percentile) of the EMG rms values during offline-scored NREM sleep. These two values marked the transition into consolidated NREM sleep in each subject during online NREM staging and were introduced to our customized SYNAPSE (TDT) project.

## Phase-targeted auditory closed-loop stimulation

For phase-targeted auditory stimulation of slow waves, SYNAPSE combines the online NREM staging feature with a phase detector. Briefly, a runtime very-narrow bandpass filter (TDT) for EEG phase detection isolated the 1 Hz component for phase targeting of slow waves (approximately 1.35 Hz in rats [*Mölle et al., 2009*]) of each subjects' left EEG channel. We predetermined slow wave's 0° as the rising zero-crossing, 90° to the positive peak, and 270° to the slow-wave trough. At every identified positive zero-crossing on the filtered signal, the phase detector resets to 0° and calculates any of the selected target phases based on the number of elapsed samples since the zero-crossing. This method offers the chance to recognize slow waves consistently across conditions, independently of the target phase. We divided the animals into three different phase-targeted stimulation approaches: up-phase stimulation targeting 60°, down-phase stimulation targeting 180°, and mock stimulation flagging slow waves at 0° (arbitrarily chosen) with no delivery of sound (*Figure 1d*). On average, a sound trigger was sent 29.30 ms (RX8 MULTI-I/O processor, TDT) upon validation of truth value for NREMratio, EMG power, and phase-target criteria. Auditory stimuli consisted of clicks of pink 1/f noise (30 ms duration, 35 dB SPL, 2 ms rising and falling slopes) in free-field conditions, from built-in speakers (MF1 multi-field magnetic speakers, TDT) on top of the stimulation chamber, 50 cm above the center of the floor area. The protocol of the mock condition was identical to up-phase or down-phase stimulations, but the sound was muted, that is, the speaker was disconnected from the RX8 MULTI-I/O processor. All triggers were time-flagged for offline analysis. Throughout all days of stimulation, we used a video system to sporadically control for stimuli-evoked arousals or reflexes.

## EEG/EMG offline scoring and online staging validation

We scored all recording files using the online computational tool SPINDLE (Sleep Phase Identification with Neural Networks for Domain-Invariant Learning) for animal sleep data (*Miladinović et al., 2019*). In short, European Data Format (.edf) files, consisting of two parietal EEG and one nuchal EMG channels, were uploaded to SPINDLE to retrieved vigilance states with 4 s epoch resolution. The algorithm classified three vigilance states: wakefulness, NREM sleep, and REM sleep. Additionally, epochs containing data outliers or signal perturbations related to environmental interference rather than changes in brain state were labeled as artefacts in wakefulness, NREM, or REM sleep. Wakefulness was defined based on high or phasic EMG activity for more than 50% of the epoch duration and low-amplitude but high-frequency EEG. NREM sleep was characterized by reduced or no EMG activity, increased EEG power in the frequency band <4 Hz, and the presence of SOs. REM sleep was defined based on high-theta power (6–9 Hz frequency band) and low-muscle tone. To validate our online NREM staging method, we contrasted all recording outputs (NREMratio ∧ EMG power) with offline scores from SPINDLE. Since online staging categorizes ongoing signal in 1 s steps, these were grouped in blocks of four in order to match the length of offline scored epochs. Online scores (i.e., blocks of four values corresponding to each 1 s online assessment) were categorized as NREM sleep every time two or more 1 s values were validated true for NREM sleep. To attest for performance, we calculated sensitivity, specificity, and precision between the

online and offline arrays. *Sensitivity* was assessed as the ratio between online-labeled NREM epochs confirmed offline as NREM sleep (true positives) and the total NREM epochs detected offline (true positives + false negatives), and reveals how strictly the pipeline identified sustained NREM sleep using the thresholds defined at BL. *Specificity* was calculated as the ration between online non-NREM epochs confirmed offline as other than NREM sleep (true negatives) and the total non-NREM epochs detected offline (true negatives + false positives), and measures how well the online tool can exclude all that it is not NREM sleep. Lastly, we evaluated *precision* as the proportion of online-labeled NREM epochs confirmed offline as NREM sleep (true positives) among all online NREM labels (true positives + false positives).

## Postprocessing of EEG

Time spent in NREM sleep was determined as an absolute number of minutes for BL and all subsequent 16 days under CLAS. We also calculated an NREM sleep fragmentation index, expressed as the number of NREM sleep bouts/total number of NREM sleep epochs, with higher index values expressing higher fragmentation. Furthermore, we extracted measures of global spectral responses in specific bandwidths by processing the left-hemisphere EEG signal with a custom MATLAB routine (ver. R2016b). Briefly, the EEG signal was resampled at 300 Hz, and multiplied with a basic Fermi window function $f(n) = \left(1 + e^{(5-n/50)}\right)^{(-1)}$, to gradually attenuate the first and last 2 s (n = 600).

Next, we filtered the signal between 0.5 and 48 Hz using low- and high-pass zero-phased equiripple FIR filters (Parks–McClellan algorithm; applied in both directions [*filtfilt*]; $order_{high}$ = 1880, $order_{low}$ = 398; –6 dB [half-amplitude] cutoff: high-pass = 0.28 Hz, low-pass = 49.12 Hz). The signal was visually inspected for regional artefacts (2 hr sliding window) not detected during automatic scoring: within scored NREM sleep, brief portions (<10 sample points at 300 Hz) of signal > ± [8× interquartile range] were reconstructed by piecewise cubic spline interpolation from neighboring points. We performed spectral analysis of consecutive 4 s epochs (FFT routine, Hamming window, 2 s overlap, resolution of 0.25 Hz) and normalized the power estimate of each frequency bin in relation to the total spectral power (0.5–30 Hz). To access potential interhemispheric asymmetries, we additionally extracted spectral density power from the EEG's right derivation for each sleep stage and calculated the perceptual change from BL. To normalize for effects related to procedural factors, we subtracted the changes observed in the mock group from the stimulated groups. In order to appreciate the longitudinal CLAS effect, we calculated the daily activity changes in each frequency band— delta (>0.5–4 Hz), theta (>4–8 Hz), alpha (>8–11 Hz), sigma (>11–16 Hz), and beta (>16–20 Hz)—as the daily mean power of NREM sleep epochs using the abovementioned digital filters ($order_{high}$ = 3758 and $order_{low}$ = 3861), normalized by the daily total power (0.5–30 Hz). Relative 24 hr delta- and sigma power were calculated in a similar fashion for M-T days and subsequently grand-averaged for that period.

## Postprocessing of sound triggers and evoked-response potentials

We analyzed the trigger distribution during pre- and motor-training days in several aspects: the daily number of triggers in wakefulness, NREM sleep and REM sleep; the ratio between the number of triggers in each sleep stage and the total number of triggers; trains of triggers as the count of any size sequences of triggers 1 s or less apart (as SOs in rats occur at a peak frequency of 1.35 Hz; *Mölle et al., 2009*); and ISI as the time between triggers (*Figure 1e*). Analysis of the evoked auditory ERPs was performed computing EEG snippets across all auditory triggers in NREM, within a 3.5 s window time-locked to the trigger onset, and a post-stimulus cutoff of 2 s. Only triggers with a post-ISI >1 s were included in this analysis. To assess the ability of the runtime paradigm to target 60° (up-phase) and 180° (down-phase), we filtered the left EEG channel between 0.5 and 2 Hz (same fashion as previous digital filters: $order_{high}$ = 3758, $order_{low}$ = 3861; –6 dB [half-amplitude] cutoff: high-pass = 0.39 Hz, low-pass = 1.91 Hz), followed by a Hilbert transform to define instantaneous phase. We averaged the instantaneous phase for all triggers delivered in NREM during the 16-day stimulation period and plotted the results in histograms with six bins of 60°. Counts were normalized to the total number of counts in each animal. Summary statistics were computed using the CircStats toolbox in MATLAB (*Berens, 2009*).

## Single-pellet reaching task protocol

We used a 15 × 40 × 30 cm (W × L × H) acrylic-glass chamber with a vertical window (1 cm wide, lower edge 5 cm above the cage floor) in the front wall. This window was covered by a motorized sliding door, connected to an inductive sensor in the back wall. The rat could open the mentioned door by poking the nose to the sensor in the back wall of the cage. The training procedure consisted of two phases: (1) habituation to the testing chamber and instrumental- or pre-training during which the rat is taught to operate the motorized door (further called P-T), and (2) reaching- or motor-training during which the rat is taught to use a single paw to obtain food pellets (further called M-T) (*Figure 1f*). In detail:

1. P-T: We food-restricted the animals for 24 hr before the first day of the protocol to ensure engagement in the task. We introduced all animals to the testing chamber for 3 days, during 60 min daily, and allowed familiarization to the cage and the positioning of the food pellets (45 mg, Bioserve Inc, Frenchtown, USA). On the fourth day and for 4 days, we started the pre-training routine by guiding all animals to the nose-poke sensor in the back of the cage, enabling the motorized door covering the front window to open. This routine was rewarded with a single-food pellet placed at the window edge, easily reached using the tongue. Each P-T session consisted of 100 trials—opening the window and retrieving a pellet—or 60 min, whichever occurred first.

2. M-T: Subsequent skilled reaching training was similar to P-T, but we placed the pellet on a pedestal (7 mm diameter), located 1.5 cm outside of the cage. In this position, the rat could only use a forelimb to retrieve the pellet (*Figure 1g*). On the first day, each animal quickly (during the first 5–10 reaches) demonstrated a forelimb preference, determining the placement side of the pedestal (left side if right forelimb was preferred and vice versa). The right forelimb was preferred in five subjects in the up-phase group (n = 8), six in the down-phase group (n = 8), and two in the mock group (n = 6). In addition, in the mock group, two subjects showed equal preference for left and right grasping, to which we placed the pedestal on the left. Overall, we rarely saw the animals using the non-preferred forelimb. A trial ended when the pellet was grasped or pushed off the pedestal, automatically closing the door within 1 s (via built-in inductive sensor on the pedestal that sensed the presence of the pellet). We scored a reaching attempt, or occasionally the last of multiple reaching attempts per pellet, as either successful (pellet is retrieved from the pedestal and eaten), failed (pellet is pushed off the pedestal), or drop-in (rat grasps the pellet but drops it during paw retraction outside or inside the cage). Each M-T session consisted of 100 attempts or 60 min, whichever occurred first, divided into five bins of 20 pellets or 12 min (*Figure 1h*).

## Single-pellet reaching task analysis

We defined SR and FR as the number of successes or fails (excluding drop-ins) out of 100 possible attempts. We calculated the intra-session change as the $\log_2$ of the fold-change of the number of successes of the last bin to the first bin of a session (–1 represents a 50% drop in successes, while 1 represents twice more successes from first bin). Rats that failed to achieve a minimum SR of 10% at M-T$_2$ or after would be excluded, but no subject met this criteria.

## Statistical analysis

We present all data as the mean ± SEM, unless stated otherwise. We performed statistical analyses using Prism 8.0 (GraphPad Inc, San Diego, CA), SPSS Statistics 26.0 (IBM Corp., Armonk, NY), R 4.0.2 (Boston, MA) (*Bates et al., 2015*; *Lenth, 2020*; *R Development Core Team, 2020*), and MATLAB R2016b and R2019a (Natick, MA). We performed power calculations (R 4.0.2, *power.anova.test*) based on human literature using the same stimulation paradigms. We expected a standard deviation of 15% for all group estimates referent to delta activity change from BL (up-phase = + 15% [*Ngo et al., 2013a*; *Papalambros et al., 2017*]; mock = 0%; down-phase = –10% [*Fattinger et al., 2017*; *Ngo et al., 2013a*]), to which we obtained a sample size n = 8 (for sigma level = 0.05 and power = 0.80). Multiple RM-ANOVA (adjusted using the Greenhouse–Geisser correction when necessary) was used to assess the influence of *time* (days, or bins in the SPRT analysis) and *condition* (up-phase vs. mock, and down-phase vs. mock) on NREM proportions, time course of delta and sigma, and SR on last protocol day. Paired Wilcoxon rank tests were applied to NREM amount relative to BL values since data did not follow a normal distribution for the plotted protocol

days. We used a one-sample Hotelling's $T^2$ test to explore whether the daily changes in all frequency bands have a mean equal to a null change (*Oja, 2010*). Differences on the time course of delta- and sigma frequencies in relation to BL were tested using multiple paired t-tests, corrected with Holm–Sidak method for multiple comparison. Phase targeting was evaluated using circular statistics from the CircStat toolbox (*Berens, 2009*). For the analysis of SPRT counts, Poisson distribution or, if dispersion was larger than expected, negative binomial distribution was fit to the data. Both models provide IRR for which the expected counts were estimated and analyzed using mixed regression models (fixed factors: days and groups; random factor: subjects) followed Tukey's multiple comparison tests. Significance for the models' fixed effects was estimated with type II Wald chi-square tests. Associations were tested using one-tailed Spearman's rank correlations, underlying established one-directional hypotheses. Unless stated otherwise: $n$ (up-phase) = 7; $n$ (mock) = 5 for electrophysiology results and $n$ (mock) = 6 for behavioral results; $n$ (down-phase) = 8. The alpha level for all statistical tests was set initially to 0.05.

## Acknowledgements

We thank Mr. Mark Hanus, Mr. Myles Billard, Dr. Clément Vitrac, and Dr. Laura Ferster for their valuable support. This project was funded by the Swiss National Science Foundation (grant numbers: 163,056 and 188790, CRB), the Clinical Research Priority Program 'Sleep and Health' of the University of Zurich (CRB), the Neuroscience Center Zurich with the patronage of Rahn & Bodmer banquiers (DN), and the Synapsis Foundation for Alzheimer Research through an earmarked donation of the Armin & Jeannine Kurz Stiftung (DN).

## Additional information

### Funding

| Funder | Grant reference number | Author |
| --- | --- | --- |
| Swiss National Science Foundation | 163056 and 188790 | Christian R Baumann |
| Synapsis Foundation for Alzheimer's Research | | Daniela Noain |
| Neuroscience Center Zurich, University of Zurich | Rahn and Bodmer donation | Daniela Noain |
| Armin and Jeannine Kurz Stiftung | | Daniela Noain |

The funders had no role in study design, data collection and interpretation, or the decision to submit the work for publication.

### Author contributions

Carlos G Moreira, Data curation, Formal analysis, Investigation, Methodology, Software, Validation, Visualization, Writing - original draft, Writing - review and editing; Christian R Baumann, Conceptualization, Funding acquisition, Investigation, Project administration, Resources, Writing - review and editing; Maurizio Scandella, Data curation, Formal analysis, Methodology, Software, Visualization, Writing - review and editing; Sergio I Nemirovsky, Data curation, Formal analysis, Methodology, Visualization, Writing - review and editing; Sven Leach, Methodology, Validation, Writing - review and editing; Reto Huber, Conceptualization, Methodology, Validation, Writing - review and editing; Daniela Noain, Conceptualization, Funding acquisition, Investigation, Project administration, Resources, Supervision, Validation, Visualization, Writing - original draft, Writing - review and editing

### Author ORCIDs

Carlos G Moreira ⓘ http://orcid.org/0000-0003-3197-3668
Christian R Baumann ⓘ http://orcid.org/0000-0003-3417-1978
Daniela Noain ⓘ http://orcid.org/0000-0001-5482-7933

### Ethics

All procedures were approved by the veterinary office of the Canton Zurich under license ZH231/2015.

### Decision letter and Author response

Decision letter https://doi.org/10.7554/eLife.68043.sa1
Author response https://doi.org/10.7554/eLife.68043.sa2

## Additional files

### Supplementary files

• Transparent reporting form

### Data availability

The .edf files containing EEG/EMG signal (BL and M-T$_{1-4}$), the corresponding labels detailing vigilance states (4s resolution), the temporal flags of the auditory triggers, and the counts from the single-pellet reaching task are publicly available in Dryad (https://doi.org/10.5061/dryad.bvq83bk99). All figures are accompanied by an Excel file containing the numerical data and statistical analyses is provided with this submission as well as in the Dryad repository.

The following dataset was generated:

| Author(s) | Year | Dataset title | Dataset URL | Database and Identifier |
|---|---|---|---|---|
| Gonçalves Moreira C, Noain D | 2021 | Closed-loop auditory stimulation method to modulate sleep slow waves and motor learning performance in rats | https://doi.org/10.5061/dryad.bvq83bk99 | Dryad Digital Repository, 10.5061/dryad.bvq83bk99 |

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
