## [Decision Letter]

**Acceptance summary:**

This manuscript provides a set of proof-of-concept data for a rodent model of closed-loop auditory stimulation during sleep as a method for augmenting NREM sleep thalamocortical oscillations. There is translational value to this approach as this has been previously used in human subjects- altering slow wave oscillation to modulate memory consolidation. Applying this tool to rodent research in future studies may allow for bridging some of the putative mechanisms underlying memory consolidation and behavioral changes observed with sleep.

**Decision letter after peer review:**

Thank you for submitting your article "Closed-loop auditory stimulation method to modulate sleep slow waves and motor learning performance in rats" for consideration by *eLife*. Your article has been reviewed by 2 peer reviewers, one of whom is a member of our Board of Reviewing Editors, and the evaluation has been overseen by Barbara Shinn-Cunningham as the Senior Editor. The reviewers have opted to remain anonymous.

Overall, the reviewers agreed that this manuscript was well suited for a Tools and Resource paper. They were enthusiastic about the contribution of this CLAS method to the field, particularly for providing a tool for future investigations into the neural mechanisms underlying sleep and behavior (e.g., cognition). However, there were several concerns list below that would need to be addressed before consideration for publication.

Essential revisions:

1) The authors have 4 EEG channels recorded from each animal. The authors state that they report only left hemisphere. It is unclear which of the two left hemisphere EEG channels is used, or whether this represents an average of both. The 4 channels are the major source of neurobiological information for the study, and provide the primary basis for assessing effects of CLAS. One can only conclude that the authors have not maximized their use of available data, and there is no clearly stated reason why. The authors should clearly describe and defend why any EEG channels were excluded. Moreover, they should provide analysis of data across the various recorded channels. For example, how does CLAS affect coherence of oscillations between the EEG sites? Are there region-specific changes in CLAS effects on oscillations? How does motor learning and handedness of task performance affect each of the 4 recording locations? All of these questions can and should be clearly addressed, using the data presently available, in the results. Having this information would make interpretation of all of the other data more straightforward.

2) The authors don't seem extremely forthright in discussing their obvious lack of effect of upstate-targeted CLAS on motor learning. They should just state this clearly in the abstract, as well as the results. The framing of the results seems unbalanced. For example, the title of Figure 7: "Up-phase stimulated animals attempt more, succeed more and fail less over time, whereas down-phase stimulated rats show a significantly reduced success-rate." There is no clear evidence that "attempts" are any different between upstate CLAS and sham CLAS animals, so this is not a fair statement (which seems to attribute attempts increasing to upstate CLAS, rather than attempts decreasing to downstate CLAS). Moreover, these differences are present on the first day of training, suggesting they are in no way related to learning. Indeed, all of the actual effects on motor task performance seem driven by deficits in the downstate-targeted rates. And all of those effects could be attributable to something very non-specific (which is not discretely tested here) such as fatigue in downstate-targeted rats. Thus I cannot agree with the way that the behavioral results are presented. Some features such as the success and failure numbers in Figure 7 are totally irrelevant in light of the attempt numbers being a variable. Success and failure rates (normalized to attempts) and changes in those rates across the multi day training are the only relevant variables. These should be measured and discussed in more concrete terms.

3) The methodology for the CLAS itself is the major novel contribution, and it would be nice to have more detailed analysis of this in the results. The early description can easily be misconstrued by readers as being targeted to NREM, however, on closer reading it is clear that the triggering can occur in any state. It would be useful to get a sense of how many (and proportion) of the CLAS triggers occur in wake and REM, and what effects these off-target triggers have on EEG activity. Is this what is being shown in 3b? It is unclear whether this is showing offline sleep scoring overall, or targeting. The targeting by day should also be shown, rather than a grand mean.

4) Regarding the accuracy of CLAS triggering reported in Figure 3, there are several points of confusion. For example, the definition of "sensitivity" (as opposed to "precision" should be outlined here, as well as in the results, readers, like myself, will have no idea what this means. In Figure 3a, it is not clear what these individual data points are, are these days of recording, or hours, or what?

5) The scoring of sleep in the various groups is inadequate. Each day should be scored, and each animal should be scored. For example, why are there only 3 data points in the sleep architecture measure in Figure 4 a-c? The sleep architecture should be shown for each and every animal, and be shown longitudinally. This would address my concern that all of the behavioral effects of CLAS (i.e., deficits in downstate CLAS rats) are driven by gradual disruption or fragmentation of NREM, or conversely by accumulated cognitive effects of disrupting NREM oscillations over days.

6) In Figure 5, there are additional points that need clarifying. I am assuming this is for offline scored NREM only? why not say this in the Figure legend? In Figure 5d-f, the choice of only pt8 (omitting pt1-7) seems a bit arbitrary to me, why not include all non-training days here?

---

## [Author Response]

Essential revisions:1) The authors have 4 EEG channels recorded from each animal. The authors state that they report only left hemisphere. It is unclear which of the two left hemisphere EEG channels is used, or whether this represents an average of both. The 4 channels are the major source of neurobiological information for the study, and provide the primary basis for assessing effects of CLAS. One can only conclude that the authors have not maximized their use of available data, and there is no clearly stated reason why. The authors should clearly describe and defend why any EEG channels were excluded. Moreover, they should provide analysis of data across the various recorded channels. For example, how does CLAS affect coherence of oscillations between the EEG sites? Are there region-specific changes in CLAS effects on oscillations? How does motor learning and handedness of task performance affect each of the 4 recording locations? All of these questions can and should be clearly addressed, using the data presently available, in the results. Having this information would make interpretation of all of the other data more straightforward.

For each rat, 2 differential EEG derivations (1 in each brain hemisphere) were recorded from a total of 4 screws implanted over the somatosensory cortex. Figure 1a illustrates both pairs of screws corresponding to left and right EEG derivations. The right EEG channel, diametrically opposed to the left one, was initially excluded from the analysis based on the assumption of a minor interhemispheric asymmetry in animals under regular sleep conditions. Based on the Reviewer request, we have now analyzed the remaining channel for protocol-day M-T3 and additionally split the subjects into right and left-pawed for separate analysis (*Derivation symmetry* in page 6 and 7, lines 145-152 of Results and Figure 4—figure supplement 1, and *handedness effect* in page 7, lines 152-163 of Results and Figure 4—figure supplement 2). We thank the Reviewer for these useful suggestions that greatly increase the robustness of our report.

2) The authors don't seem extremely forthright in discussing their obvious lack of effect of upstate-targeted CLAS on motor learning. They should just state this clearly in the abstract, as well as the results. The framing of the results seems unbalanced. For example, the title of Figure 7: "Up-phase stimulated animals attempt more, succeed more and fail less over time, whereas down-phase stimulated rats show a significantly reduced success-rate." There is no clear evidence that "attempts" are any different between upstate CLAS and sham CLAS animals, so this is not a fair statement (which seems to attribute attempts increasing to upstate CLAS, rather than attempts decreasing to downstate CLAS). Moreover, these differences are present on the first day of training, suggesting they are in no way related to learning. Indeed, all of the actual effects on motor task performance seem driven by deficits in the downstate-targeted rates. And all of those effects could be attributable to something very non-specific (which is not discretely tested here) such as fatigue in downstate-targeted rats. Thus I cannot agree with the way that the behavioral results are presented. Some features such as the success and failure numbers in Figure 7 are totally irrelevant in light of the attempt numbers being a variable. Success and failure rates (normalized to attempts) and changes in those rates across the multi day training are the only relevant variables. These should be measured and discussed in more concrete terms.

Fair point. We have down-toned the claim in relation to the behavioral results appropriately as suggested by the Reviewer. We also included more detailed statistical assessments for this task, and revised and adapted its general presentation to address the Reviewer’s concerns (Abstract: page 2, line 37; Results: pages 10 and 11, lines 232-261 and new Figure 7; Discussion: page 15, lines 359-360, 370-371; Materials and methods: page 26, lines 649-650; Figure legend: page 34, lines 831847).

3) The methodology for the CLAS itself is the major novel contribution, and it would be nice to have more detailed analysis of this in the results. The early description can easily be misconstrued by readers as being targeted to NREM, however, on closer reading it is clear that the triggering can occur in any state. It would be useful to get a sense of how many (and proportion) of the CLAS triggers occur in wake and REM, and what effects these off-target triggers have on EEG activity. Is this what is being shown in 3b? It is unclear whether this is showing offline sleep scoring overall, or targeting. The targeting by day should also be shown, rather than a grand mean.

Figure 3a represented the performance of the online staging tool and not the triggers stage distribution, i.e. we originally analyzed only the triggers flagged during NREM. Indeed, this more broad analysis was initially skipped and we judge prudent to include it now, as pointed out by the Reviewer, auditory triggers also occurred during offline-scored REM and WAKE periods. This data was integrated now in Figure 3—figure supplement 1, including the number and proportion of triggers in each stage for up- and down-state stimulation groups, grand averaged and per protocol-day (Results: page 5, lines 102-109; Materials and methods: page 24, lines 601-603).

In Figure 3b we analyze the precision (proportion of NREM sleep epochs correctly identified online) per group, in order to visualize the distribution of epochs mislabeled online across stages.

The existing figure legend was adapted for enhanced clarity and a figure legend added for the new data (page 30 and 31, lines 740-744 -existing legend- and 746-753 -new legend-).

4) Regarding the accuracy of CLAS triggering reported in Figure 3, there are several points of confusion. For example, the definition of "sensitivity" (as opposed to "precision" should be outlined here, as well as in the results, readers, like myself, will have no idea what this means. In Figure 3a, it is not clear what these individual data points are, are these days of recording, or hours, or what?

The definitions of sensitivity, specificity and precision regarding the performance of our online staging method, not to be confused with the triggers targeting, were added to the Materials and methods section (page 22 and 23, lines 556-569). In Figure 3a, the data points represent days of recording (12 days) from all stimulated animals (up- (n = 7) and down-state (n = 8) groups), more specifically, (7+8) * 12 data points. The figure legend was adapted for increased clarity (page 30, lines 740-744).

5) The scoring of sleep in the various groups is inadequate. Each day should be scored, and each animal should be scored. For example, why are there only 3 data points in the sleep architecture measure in Figure 4 a-c? The sleep architecture should be shown for each and every animal, and be shown longitudinally. This would address my concern that all of the behavioral effects of CLAS (i.e., deficits in downstate CLAS rats) are driven by gradual disruption or fragmentation of NREM, or conversely by accumulated cognitive effects of disrupting NREM oscillations over days.

Point well taken. Initially, we decided to only show the most meaningful protocol-days related to the training routine of the single-pellet reaching task. To meet the Reviewer’s concern, we now present sleep architecture longitudinally (Results: page 6, lines 121-130 and new Figure 4a, b; Figure legends: page 31, lines 755-761).

6) In Figure 5, there are additional points that need clarifying. I am assuming this is for offline scored NREM only? why not say this in the Figure legend? In Figure 5d-f, the choice of only pt8 (omitting pt1-7) seems a bit arbitrary to me, why not include all non-training days here?

Yes, the data presented in this figure corresponds indeed to off-line scored NREM only. We have now clarified this further in the figure legend (page 32, line 787-788).

In relation to Figure 5d-f, the choice of presenting data from protocol-day P-T8 was not arbitrary. We included day P-T8 along with the MT1-4 in the grand-average of the motor-training phase with the intention to always show δ-activity 1 day before and after each SPRT session. This objective related to the well-accepted notion that not only δ activity may change the performance on a behavioral test but also the training/testing has an effect on following SWA, therefore, making a point of interest evaluating both time windows. This was now more explicitly explained in the figure legend (page 32, lines 794-797).